# Plasma-assisted manipulation of vanadia nanoclusters for efficient selective catalytic reduction of NO$_x$

Yong Yin[1,6], Bingcheng Luo[2,6], Kezhi Li[3], Benjamin M. Moskowitz[4], Bar Mosevitzky Lis[4], Israel E. Wachs[4] ✉, Minghui Zhu[5] ✉, Ye Sun[1], Tianle Zhu[1] & Xiang Li[1] ✉

Supported nanoclusters (SNCs) with distinct geometric and electronic structures have garnered significant attention in the field of heterogeneous catalysis. However, their directed synthesis remains a challenge due to limited efficient approaches. This study presents a plasma-assisted treatment strategy to achieve supported metal oxide nanoclusters from a rapid transformation of monomeric dispersed metal oxides. As a case study, oligomeric vanadia-dominated surface sites were derived from the classic supported V$_2$O$_5$-WO$_3$/TiO$_2$ (VWT) catalyst and showed nearly an order of magnitude increase in turnover frequency (TOF) value via an H$_2$-plasma treatment for selective catalytic reduction of NO with NH$_3$. Such oligomeric surface VO$_x$ sites were not only successfully observed and firstly distinguished from WO$_x$ and TiO$_2$ by advanced electron microscopy, but also facilitated the generation of surface amide and nitrates intermediates that enable barrier-less steps in the SCR reaction as observed by modulation excitation spectroscopy technologies and predicted DFT calculations.

Solid catalysts are employed in the production of over 80% of chemicals on a global scale[1]. Atomically dispersed supported solid catalysts, including single-atom catalysts (SACs) and multi-atom cluster catalysts, have recently garnered significant attention because of the maximum atom utilization, optimized charge distribution, and tuned coordination environment[2–5]. The SACs possess well-defined active centers and a unique confinement effect, while they may not be universally applicable to reactions that require multinuclear or adjacent active sites[6–8]. Beyond a simple combination of SACs, multi-atom cluster catalysts could result in enhanced activity because of the synergistic effects between adjacent atoms[8–10]. Such nanocatalysts have demonstrated remarkable catalytic performance in various reactions, such as CO oxidation, selective oxidation of hydrocarbons, selective catalytic reduction, selective hydrogenation, and electrochemical CO$_2$ reduction[8–10]. However, achieving precise control over smaller oligomeric clusters (i.e., metal-oxo or metal cluster) is notably challenging, as they are highly susceptible to undergoing Oswald ripening, resulting in the gradual enlargement of these smaller clusters and formation of nanoparticles[11–14].

Selective catalytic reduction of NO$_x$ with NH$_3$ (i.e., NH$_3$-SCR) to benign N$_2$ and H$_2$O reaction products by supported vanadia-based catalysts has been widely applied to control NO$_x$ emission from coal- and natural gas-fired power plants[15–19]. It is found that the active moieties of the supported vanadia-based catalysts are largely determined by the dispersed vanadyl surface sites[20,21]. Oligomeric surface vanadyl sites (dimers, trimers, etc.) have been proposed to hold higher intrinsic activity

[1]School of Space and Environment, Beihang University, Beijing 100191, China. [2]College of Science, China Agricultural University, Beijing 100083, China. [3]Institute of Engineering Technology, Sinopec Catalyst Co. Ltd., Beijing 101111, China. [4]Operando Molecular Spectroscopy & Catalysis Laboratory, Department of Chemical and Biomolecular Engineering, Lehigh University, Bethlehem, PA 18015, USA. [5]State Key Laboratory of Chemical Engineering, East China University of Science and Technology, 130 Meilong Road, Shanghai 200237, China. [6]These authors contributed equally: Yong Yin, Bingcheng Luo. ✉e-mail: iew0@lehigh.edu; minghuizhu@ecust.edu.cn; xiangli@buaa.edu.cn

than isolated vanadyls sites at low temperatures for this bimolecular reaction[22–24]. Recently, the achievement of predominantly oligomeric vanadia surface sites for supported SCR catalysts was shown to be regulated through the loading amount of the active surface $VO_x$ or the surface $WO_x$ promoter prepared by incipient-wetness impregnation. With careful control of the vanadia loading below monolayer surface coverage on the anatase support (-0.1 wt% $V_2O_5$/m² $TiO_2$ or ~8 V atoms/nm²), the predominant surface $VO_x$ sites are found to be monomeric al low surface coverage (<0.03 wt% $V_2O_5$/m² $TiO_2$) and oligomeric at high surface coverage (0.05-0.1 wt% $V_2O_5$/m² $TiO_2$)[20,22,25–27]. However, as a case of the commercial $TiO_2$ P25 support (56 m²/g), achieving the predominant formation of oligomeric vanadia surface sites requires a vanadia loading of ~3.5 wt% V (5 wt% $V_2O_5$), which would cause a serious degradation of the $N_2$ selectivity and potential biological toxicity from volatilization of some vanadia[27,28]. Hence, it remains a significant challenge to precisely regulate a high concentration of vanadia clusters for commercially supported vanadia-based catalysts at low contents of active components. Despite extensive research recognizing the pivotal role of nitrates in the $NH_3$-SCR reaction, the current study falls short in elucidating the correlation between nitrate formation and catalyst structure and lacks atomic-level insights into how nitrates participate in the reaction. There is a pressing need to address this issue and provide atomic-level insights into how nitrates engage in the $NH_3$-SCR reaction[29–34].

As illustrated in Fig. 1a, in the present study, a novel approach for the synthesis of supported vanadyl nanoclusters is presented through a secondary-level $H_2$ plasma modification of conventional supported vanadia-based catalysts. Structural characterization revealed the transformation of isolated surface vanadyl sites into vanadia nanoclusters on the modified $TiO_2$ surface during the plasma treatment. The resulting catalyst exhibited remarkable catalytic activity and stability in the $NH_3$-SCR at low temperatures, with a tenfold increase in the turnover frequency (TOF). With the capabilities of advanced scanning transmission electron microscopy (STEM) and modulation excitation spectroscopy (MES) technologies as well as density functional theory (DFT) theoretical calculations, the distribution of supported vanadia clusters was identified at the atomic scale and the $NH_3$-SCR reaction pathway was found to involve surface nitrate reaction intermediates.

## Results and discussion
### Preparation and identification of surface $VO_x$ sites on $TiO_2$
The originally supported $V_2O_5$–$WO_3$/$TiO_2$ catalysts, "OR" for short, were synthesized by the incipient-wetness impregnation-drying-

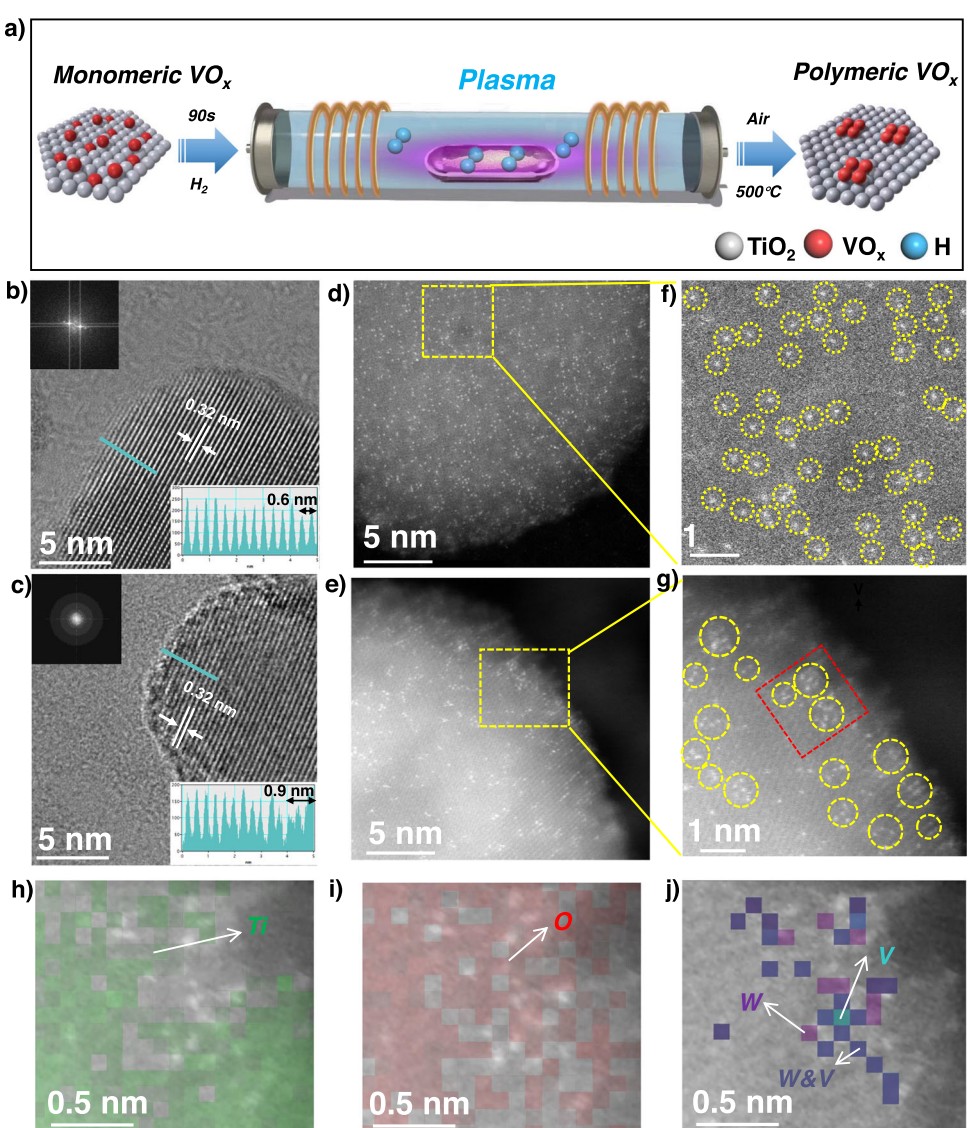

**Fig. 1 | Synthesis and electron microscopy of the samples. a** Schematic of the surface vanadyl species under plasma treatment over vanadia-based catalysts. HRTEM images of (**b**) OR and **c** PL. Selected high-angle annular dark field (HAADF) images of (**d**) OR and (**e**) PL. Enlarged view of the yellow region in **d**, (**f**) and **e**, **g**. 2D atomic maps of the EELS signals of (**h**) Ti, (**i**) O, and **j** V and W in combination with the simultaneously acquired HAADF image of the red region in (**g**).

calcination method with a pure $TiO_2$ (anatase) support. The OR powder was then subjected to an $H_2$ plasma treatment using a radio-frequency discharge source within a Plasma Enhanced Chemical Vapor Deposition (PECVD) system. The plasma-modified catalysts, referred to as "PL" catalysts, were obtained after air calcination at 500 °C (Fig. S1). The loadings of V and W were quantified by inductively coupled plasma mass spectrometry (ICP-MS, Table S1). In the OR sample, the V content was found to be 0.81 wt% and the W content was 3.16 wt%. Similarly, in the PL sample, the V content was measured at 0.80 wt% and the W content at 3.18 wt%. X-ray diffraction (XRD) analysis of both the OR and PL catalysts exhibited diffraction peaks solely attributed to the anatase phase of the $TiO_2$ support, with no discernible XRD peaks from crystalline $V_2O_5$ and $WO_3$ nanoparticles (Fig. S2). The size distribution and morphology of the supported $V_2O_5$–$WO_3$/$TiO_2$ catalysts were examined using high-resolution transmission electron microscopy (HR-TEM) (Fig. S3). The plasma treatment did not result in significant changes in the average particle size (~21 nm) or interplanar spacing (0.32 nm) of the titania support compared to the pristine $TiO_2$ (anatase) (Fig. 1b, c). For the PL catalyst (90 s plasma treatment), however, a notable lattice distortion resembling a core-shell structure of ~0.9 nm thickness of the shell as expected for amorphous V–W–O monolayer, larger than OR (~0.6 nm), along with blurred diffraction patterns at the edges is observed[35]. Prolonged plasma treatment (300 s) resulted in a further increased thickness of the distorted layer to 1.4 nm (Fig. S4).

The dispersion of the surface atoms was further examined by spherical aberration-corrected high-angle annular dark-field scanning transmission electron microscopy (HAADF-STEM). In the case of the OR catalyst (Fig. 1d, f), isolated atoms labeled within the indicated yellow circles are observed, indicating the presence of uniformly dispersed single atoms on the $TiO_2$ support. In contrast, for the corresponding PL catalyst (Fig. 1e, g), fully exposed non-crystalline island-like nanoclusters, smaller than 0.8 nm in diameter, are observed. These agglomerated bright spots are attributed to W atoms given their significantly larger atomic number ($Z = 74$) compared to V ($Z = 23$) and Ti ($Z = 22$). To further differentiate between V and Ti in the PL catalyst, given their close $Z$ numbers, a spectrum imaging technique was employed within a selected region (red region in Fig. 1g) to investigate V atoms within the nanoclusters. Electron energy loss spectroscopy (EELS) mapping within the electronic energy loss range of 350–850 eV was acquired (Fig. S5). The selected regions displayed a uniform distribution of the Ti element without any bright spots on the surface (Fig. 1h, i). Interestingly, the EELS mapping of V atoms precisely localized to the bright spots (Fig. 1j), indicating that V and W occupy the same positions within the nanoclusters. All the above confirmed that surface vanadyl sites on the surface of the $TiO_2$ support are co-located with surface tungsten sites.

The short-range structure of the reactive dehydrated surface $VO_x$ sites was further investigated with spectroscopic techniques. For the OR catalyst (Fig. 2c), characteristic Raman bands at 1005 and 1023 cm⁻¹ correspond to the vibrations of terminal W=O and V=O bonds, respectively[36–39]. For the PL catalyst, the Raman band of the V=O vibration shifted from 1023 to 1030 cm⁻¹, indicating an increase of oligomerized degree of the surface $VO_x$ sites[24,25]. The PL catalyst also

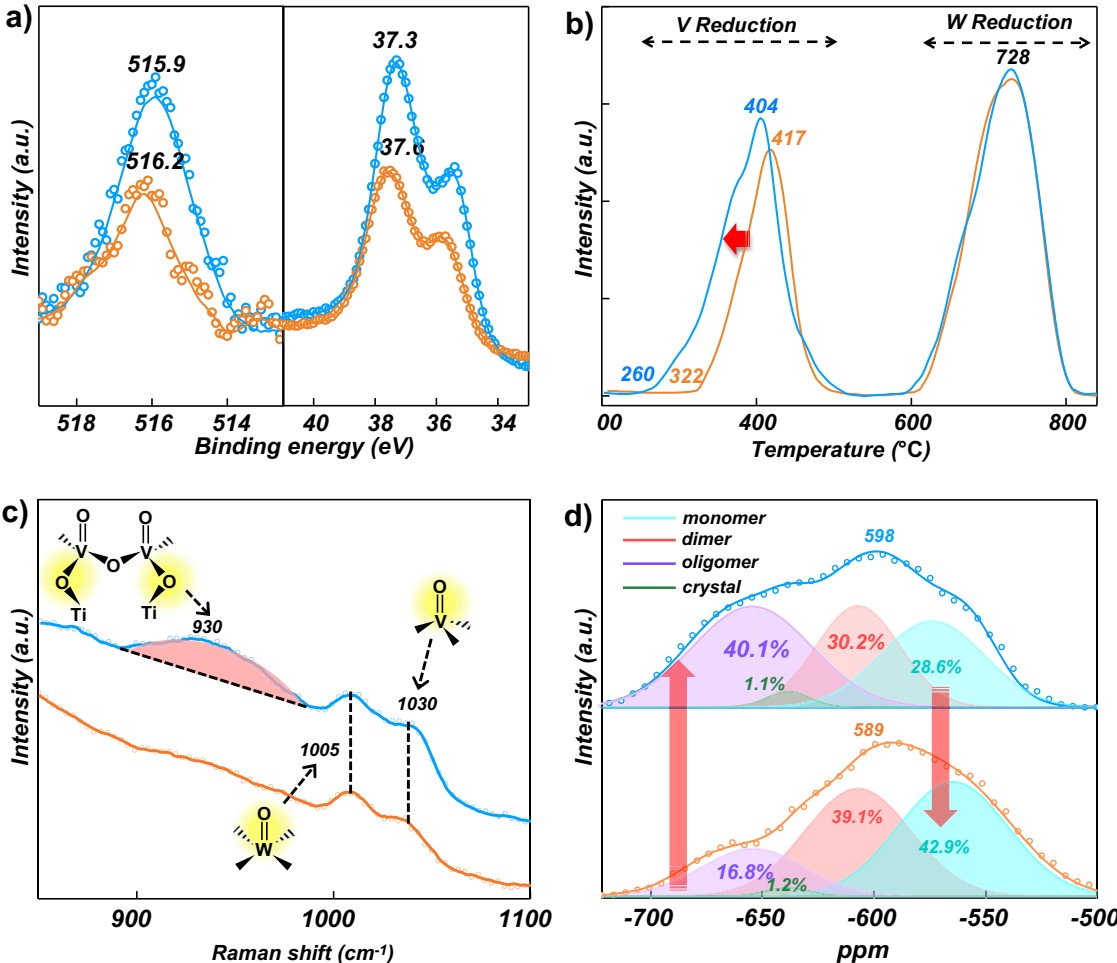

**Fig. 2 | Spectral characterization of the samples. a** V2*p* and W4*f* XPS spectra, **b** $H_2$-TPR profiles of PL (blue) and OR (red). **c** In situ dehydrated Raman spectra, and **d** deconvolution of the in situ solid-state ⁵¹V MAS NMR spectra of PL (blue) and OR (red).

exhibited a broad band centered at ~930 cm$^{-1}$ assigned to the V–O vibration from the bridging V–O–Ti bond[27]. The lack of shift of the Raman band for the W=O bond for the PL catalyst suggests that the surface WO$_x$ species in the OR catalyst were minimally affected by the PL treatment. The absence of a crystalline V$_2$O$_5$ band at ~995 cm$^{-1}$ confirmed that V$_2$O$_5$ NPs were not present[39]. Complementary supporting information about the states of the surface VO$_x$ species was provided by solid-state $^{51}$V MAS NMR spectroscopy in Fig. 2d. The $^{51}$V MAS NMR curves were deconvoluted and fitted into four sub-peaks of distinct vanadyl species with different extents of oligomerization (i.e., monomer, dimer, oligomer (trimer and longer) and crystalline V$_2$O$_5$ nanoparticles)[40–42]. The crystalline vanadyl sites were most probably related to surface VO$_x$ sites that had a similar structure because the crystalline V$_2$O$_5$ Raman band was not present at 995 cm$^{-1}$ and would give a strong Raman band. An increase of oligomeric surface VO$_x$ sites (peak at −652 ppm) from 15 to 41% was found with the plasma treatment and the monomeric surface VO$_x$ sites (peak at −567 ppm) decreased from 57 to 24%. Therefore, a catalyst with predominantly oligomeric surface VO$_x$ species on the TiO$_2$ support was successfully prepared through the applied "top–down" plasma treatment approach[43].

The surface contents of V and W of the PL (1.25 at% and 6.21 at%) were significantly higher than the OR (0.91 at% and 4.0 at%) in the outer surface region (1–3 nm) from X-ray photoelectron spectroscopy (XPS) results, respectively (Fig. 2a, Table S1, and Fig. S6). The H$_2$-TPR results reveal the easier reduction of surface V oxides while almost no change for W oxides. This was attributed to the stronger reducibility of surface-enriched V, indicating that the former was enriched and aggregated in the distorted layer (Fig. 2b)[39]. The above findings indicate that the active components, especially some of the dispersed VO$_x$ species, migrated to the topmost surface region (~1–3 nm) after the plasma treatment. The High Sensitivity-Low Energy Ion Scattering (HS-LEIS) results also supported the XPS findings (Figs. S7 and S8). For the

OR catalyst, the V (3 atomic %) signal was in the minority compared to the W (39 atomic %) and Ti (58 atomic %) signals (Fig. S7). In contrast, the topmost surface region of the PL catalyst contains a much larger amount of V atoms, and the percentage of V reaches 28 atomic % with the contents of W and Ti decreasing to 52 atomic % and 28 atomic %, respectively (Fig. S8).

## NO/NH$_3$-SCR performance of supported VO$_x$–WO$_x$/TiO$_2$ catalysts

The effect of plasma treatment time on the NO/NH$_3$-SCR activity of the supported V$_2$O$_5$–WO$_3$/TiO$_2$ catalysts was investigated under a high gas hourly space velocity (GHSV) of 375,000 cm$^3$/(g h) in a fixed-bed reactor (Fig. S9). The NO reaction rate at 200 °C for the OR catalyst was $0.24 \pm 0.02 \times 10^{-6}$ mol g$^{-1}$ s$^{-1}$. After plasma treatment times of 60 s and 90 s, the reaction rate increased to $1.3 \pm 0.03 \times 10^{-6}$ mol g$^{-1}$ s$^{-1}$ and $2.8 \pm 0.04 \times 10^{-6}$ mol g$^{-1}$ s$^{-1}$, respectively (Fig. 3a). With longer treatment times, however, the reaction rate decreased to $2.3 \pm 0.04 \times 10^{-6}$ mol g$^{-1}$ s$^{-1}$ (120 s) to $1.1 \pm 0.03 \times 10^{-6}$ mol g$^{-1}$ s$^{-1}$ (300 s). The N$_2$ selectivity, however, showed negligible change and was always around 99%. The increasing disparity in NO reaction rates between the OR and PL (90 s) catalysts as the reaction temperature rises is illustrated in Fig. 3b. The apparent activation energy ($E_a$) of the PL (90 s) catalyst was $27.9 \pm 1.8$ kJ/mol, which was significantly lower to the OR catalyst with an $E_a$ of $41.6 \pm 2.3$ kJ/mol.

Multi-cycle and long-term performance tests were conducted to assess the catalytic stability of the PL catalyst. Five consecutive cycles were performed, and the NO conversions and N$_2$ selectivity remained unaltered throughout the cycles (Fig. 3c). Additionally, a 120-h catalytic stability test showed that the PL catalyst maintained a stable NO conversion of 97.1% at 270 °C. In contrast, the OR catalyst exhibited a lower NO conversion of 72.0% under similar conditions (Fig. S12). The TOF value of PL was superior to the commercial or reported SCR catalysts at the same or higher WHSV at 200 °C (Fig. 3d)[23]. Moreover, it

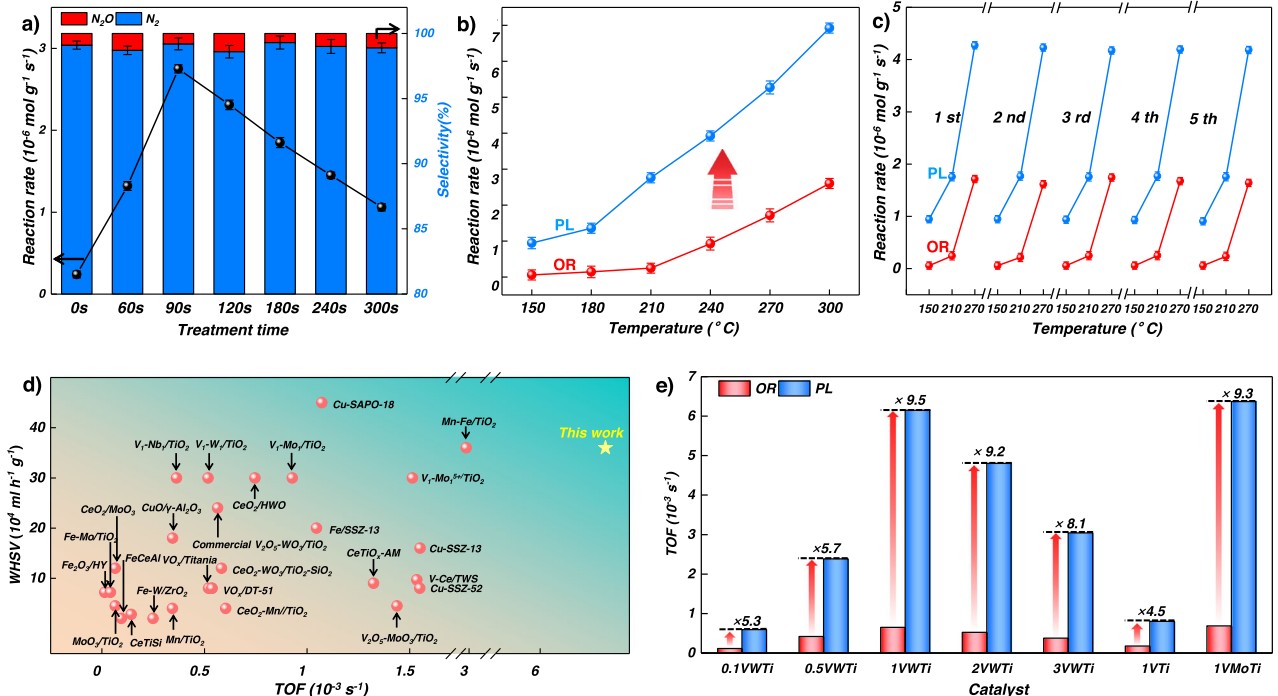

**Fig. 3 | Catalytic performance of PL and OR catalyst in the NH$_3$-SCR. a** Reaction rate and N$_2$ selectivity of PL with different treatment time at 200 °C for NH$_3$-SCR. **b** Reaction rate and apparent activating energy ($E_a$,inset) of PL and OR. **c** NO/NH$_3$-SCR cycle stability of OR (orange) and PL (blue). Comparison of TOF values for the **(d)** reported NO/NH$_3$-SCR catalysts and **(e)** different vanadia-based (PL and OR) NO/NH$_3$-SCR catalysts at 200 °C. Error bars are standard deviations were calculated from triple activity testing.

was found that all supported $V_2O_5/TiO_2$, $V_2O_5-MoO_3/TiO_2$, and $V_2O_5-WO_3/TiO_2$ catalysts with a different V loading (0.1–3.0 wt%) and stable W and Mo loading (3.22 ± 0.1) (Table S2) exhibited 4.5- to 9.5-fold increases in TOF values after a plasma treatment (Fig. 3e), indicating our strategy is universal for SCR activity improvement of supported vanadia-based catalysts.

## Exploration of plasma effect on catalyst

In order to investigate the plasma effect on surface defects, we firstly prepared pure $TiO_2$ samples with varying plasma treatment durations for reference. It was found that the $E_g$ value of Ti slightly increased from 3.2 to 3.5 eV (Fig. S13 and Table S3) from UV–Vis spectroscopy, meanwhile the $B_{1g}$, $A_{1g}$, and $E_g$ signals (at 399, 518, and 641 $cm^{-1}$) belonging to $TiO_2$ (anatase) decreased in intensity from Raman spectroscopy (Fig. S14) with increasing treatment duration. This indicates generation of defect sites on the $TiO_2$ particles from the plasma treatment[44]. However, these signals corresponding to $TiO_2$ defect sites disappear after calcination (Fig. S13 and Table S3), suggesting that defect sites could be filled by $O_2$ at high temperatures that may diminish the enhanced catalytic activity for PL. To further substantiate this hypothesis, three controlled samples were also prepared by impregnation of vanadia precursors (ammonium metavanadate) on plasma treated $TiO_2$ ($V_2O_5-WO_3(P)/TiO_2$), simultaneous impregnation of the vanadia and tungsta (aqueous ammonium tungstate) precursors on plasma treated $TiO_2$ ($V_2O_5-WO_3/TiO_2(P)$), and impregnation of the aqueous ammonium metavanadate precursor on plasma treated $WO_3/TiO_2$ and dried at 100 °C and then calcined at 500 °C in air ($V_2O_5-WO_3/TiO_2(RC)$). The significant Raman bands associated with the $TiO_2$ (anatase) and V=O vibrations of these catalysts remained almost unchanged (Fig. S14). However, all of these catalysts possessed poorer activity than untreated supported $V_2O_5/TiO_2$, $V_2O_5-WO_3/TiO_2$, or PL catalysts (Fig. S15). This suggests that the improved performance of the PL catalysts is not related to surface defects on the support, but rather to the interaction of $TiO_2$ with the surface vanadia species during the plasma chemical process.

To identify the function of $H_2$, a series of extended experiments with different plasma treatment atmospheres (Ar and $O_2$) and $H_2$ thermal treatment without plasma were undertaken with the PL catalyst. The results suggested that only the Ar plasma treatment presented a slight activity enhancement (Fig. S10), while $H_2$ thermal treatment had a negative effect on the SCR performance, especially at high temperatures (Fig. S11). These findings, therefore, further corroborate that an appropriate reductive atmosphere was significant for surface $VO_x$ cluster formation and activity improvement.

As plasma interacts with the catalyst surface, a considerable quantity of particles is projected onto the material. During this interaction, involving ions, neutral particles, and the material surface, the kinetic energy of the incident particles is transferred to surface atoms via collision cascades[44–50]. Given that the bond strength of Ti–O (dissociation enthalpy of 662 kJ/mol) is higher than that of V–O (644 kJ/mol)[51]. Additionally, the high coordination number (CN = 6) and the octahedral structure of titanium dioxide contribute to the enhanced stability of the crystal structure. When the absorption energy of the V–O bonds in the plasma exceeds their inherent bond energy, the bonds become increasingly prone to disruption, leading to the migration of V atoms to the catalyst's surface. Here, V atoms aggregate, bonding to form polymeric vanadium oxide. This aggregation effectively lowers the system's energy[52], highlighting a crucial aspect of the catalyst's interaction with plasma.

## Investigation on the SCR reaction mechanism

*Operando* DRIFTS spectra during temperature-programmed measurements were initially undertaken with an online MS detector at the reaction cell outlet under the $NO-NH_3-O_2-Ar$ reaction mixture from 100 to 300 °C. The OR catalyst showed characteristic IR peaks

attributed to the surface $NH_4^+$ species adsorbed on Brønsted acid sites ($B-NH_4^+$: 1400 and 1670 $cm^{-1}$), surface $NH_3$ coordinated at Lewis acid sites ($L-NH_3$: 1230 and 1604 $cm^{-1}$) and adsorbed $NO_2$ at 1340 $cm^{-1}$ as shown in Fig. S16a[16,21,22]. In the case of the PL catalyst, at 100 °C, $L-NH_3$ (at 1604 $cm^{-1}$) and adsorbed $NO_2$ (at 1340 $cm^{-1}$) are also observed. As the temperature increased to 150–250 °C, additional infrared (IR) peaks appeared, including from surface bridging nitrate ($\nu_s$ $(N-O)_2$ at 1288 $cm^{-1}$ and $\nu_{as}$ $(N-O)_2$ at 1599 $cm^{-1}$), bidentate nitrate ($\nu_{as}$ $(N-O)_2$ at 1579 $cm^{-1}$), and $NH_2NO$ ($\nu_s$ (N–H) at 1330 $cm^{-1}$, $\nu$ (N=O) at 1490 $cm^{-1}$) as shown in Fig. S16b[18,53–57]. For the PL catalyst, the temperature for formation of $N_2$ in the outlet initiated at 115 °C that was much lower than the initiation temperature of 180 °C for the OR catalyst reflecting the greater activity for the PL catalyst (Fig. S17).

The MES-DRIFTS measurements were conducted at 150 °C in order to determine the participating surface species in the SCR reaction. The MES studies employed alternating pulses of $NH_3$ and NO while maintaining a constant $O_2$ concentration (5 vol%) in a flowing Ar environment (Details given in Figs. S18, S20, and S22). In Fig. 4a and b, the PL catalyst exhibited the MES-DRIFTS peaks from $NH_3$ related peaks (1190, 1370, 1460, 1542, 1618, 3260, 3406 $cm^{-1}$) and adsorbed $H_2O$ (1618 $cm^{-1}$). The V=O (2035 $cm^{-1}$) at overtone region showed an opposite sign to $NH_3$ introduction, indicating $NH_3$ adsorption on it (Fig. S19). The bridging nitrates (1288 $\nu_s(N-O)_2$ and 1599 $cm^{-1}$ $\nu_{as}(N-O)_2$), bidentate nitrates (1260 $\nu_s(N-O)_2$ and 1579 $cm^{-1}$ $\nu_{as}(N-O)_2$) and bridging M–O(H)–M (3653 $cm^{-1}$) respond to NO introduction[21,58,59]. The phase delay for bridging M–(OH)–M is opposite in phase to the surface $NH_3^*$, $NH_4^{+*}$, and $NH_2^*$, suggesting that N–H cleavage does form terminal V–OH hydroxyls rather than bridging hydroxyls.

Due to the overlap of IR peaks associated with bidentate nitrate, bridging nitrate, $L-NH_3$, and $H_2O$ around 1600 $cm^{-1}$, there is mutual interference among the IR peaks of these species. To address this challenge, MES-DRIFTS studies utilized isotopically labeled reactant $ND_3$ to prevent overlapping[60–63]. As shown in Fig. 4c, the IR peaks of adsorbed $ND_2$ (1124 $cm^{-1}$), nitrate intermediates (1260, 1288, 1579, and 1599 $cm^{-1}$), and $D_2O$ (1385 $cm^{-1}$) were still observed for the PL catalyst. In contrast to Marberger et al.'s study, isotope experiment confirmed that the band at 1599 $cm^{-1}$ should be attributed to nitrate rather than $NH_3$, suggesting the involvement of nitrate in the reaction[54]. The assignment for the aforementioned peaks has been reinforced based on Marberger et al., with a detailed comparison provided in Section 7 of the Supplementary Information[54]. The coverage of bridging bidentate $NO_3^*$ increases substantially in this experiment compared to the $NH_3$ experiment, by using the hydroxyl mode as an internal reference. It is proposed that surface $NH_3^*$ ($ND_3^*$) undergoes N–H (N–D) cleavage to yield surface $NH_2^*$ ($ND_2^*$) that then associatively couples with surface $NO_3^*$ to yield the surface $NH_2NO_2(OH)$ intermediate. Given that the N–D cleavage is kinetically slower than N–H cleavage, this increases the surface coverage of $NO_3^*$ that is not consumed as quickly. This might close the reduction half cycle if further N–H bond breaking occurs to release $H_2O$ and $N_2$. In comparison, the characteristic bands belonging to adsorbed nitrate and amide species were not found for the OR catalyst in the MES-DRIFTS (Figs. S21a and S23).

The evolution of the normalized signal intensity of surface intermediate species appearing in Fig. 4b as a function of phase angle were examined (Fig. S24) to further discern the chronological order of the species' appearance. The phase angle of $L-NH_3$ precedes that of $B-NH_4^+$ by 10°, indicating faster reactivity of $L-NH_3$ than $B-NH_4^+$. Previously, in situ time-resolved IR spectroscopy demonstrated that $L-NH_3$ would not convert to $B-NH_4^+$ at below 200 °C during $NH_3$-SCR[64]. Therefore, surface $L-NH_3$ appears to be the primary active site and species involved in the SCR reaction in this study. In contrast, the peaks of surface $NH_2$ shifted by 20° with respect to the $L-NH_3$ peaks. Simultaneously, the IR peak from surface $NH_2NO$ species displays a phase shift of 10° relative to the peaks from the surface $Bi-NO_3$ and $Bri-NO_3$ species, suggesting a temporal delay in the formation of

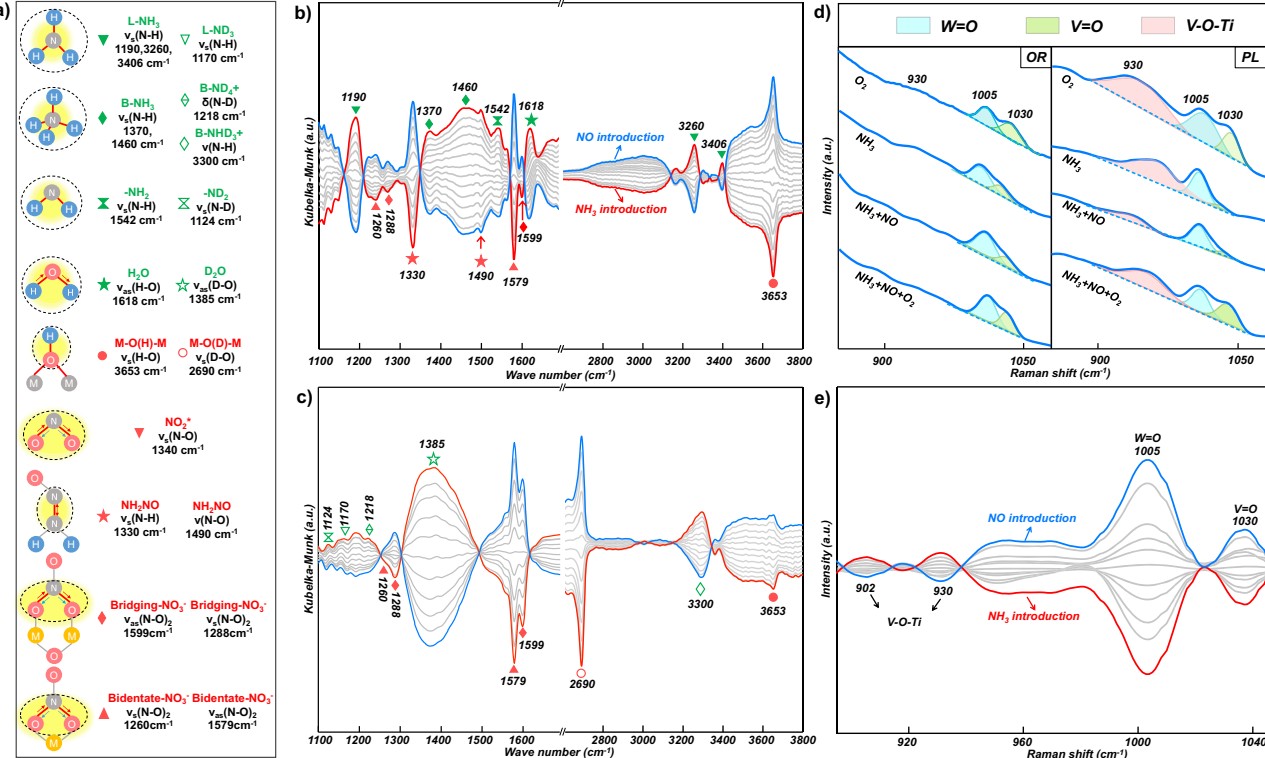

**Fig. 4 | MES DRIFT, Raman, and in-situ Raman spectra on PL and OR catalyst. a** Schematic of surface species and their corresponding IR vibrations (The yellow shading represents the positions of IR vibrations). MES DRIFT spectra of PL during (**b**) NO + O$_2$/NH$_3$ + O$_2$ and (**c**) NO + O$_2$/ND$_3$ + O$_2$ modulation experiment. **d** In situ Raman spectra of OR and PL under different reaction conditions (5% O$_2$/Ar,

2000 ppm of NH$_3$/Ar, 2000 ppm of NO + 2000 ppm of NH$_3$/Ar, 2000 ppm of NO + 2000 ppm of NH$_3$ + 5% O$_2$/Ar in sequence). **e** MES-Raman spectra of PL during NO + O$_2$/NH$_3$ + O$_2$ modulation experiment. The above experiments were carried out at 150 °C.

surface NH$_2$NO species with respect to the generation of surface nitrate species. The appearance of NH$_2$* is delayed even more (20°) with respect to the introduction of NH$_3$ than the appearance of surface NH$_2$NO* (10°) upon the introduction of NO*, this indicates that N–H cleavage as rate-determining step. Hence, it appears that the PL catalyst exhibits a distinct reaction pathway involving surface L–NH$_3$ and adsorbed nitrate species, potentially leading to the formation of a surface NH$_2$NO intermediate for SCR[65–70].

In situ, Raman experiments were performed to investigate the molecular structures of the catalytic active sites involved in the nitrate route. For both the OR and PL catalysts under an O$_2$ environment, terminal V=O (1023–1030 cm$^{-1}$), W=O (1005 cm$^{-1}$), and bridging V–O–Ti (-930 cm$^{-1}$) vibrational bands are present with the bridging V–O–Ti vibration very strong for the PL catalyst (Fig. 4d)[27,36]. Upon introduction of NH$_3$, the V=O band selectively diminishes reflecting the interaction of ammonia with this bond. Upon the addition of NO to the NH$_3$ stream, the bridging V–O–Ti band selectively decreases, especially pronounced for the PL catalyst, suggesting a correlation between the changes in the bridging V–O–Ti bond and the reaction of NO. When O$_2$ is added to the NH$_3$ + NO stream, the V=O, and V–O–Ti bands increase in intensity because of the oxidation of the reduced surface VO$_x$ sites and consumption of the surface ammonia species that broaden the IR bands. The intensity of the W=O band is less dramatically affected by the changing environmental conditions reflecting its inability to undergo efficient redox and ammonia coordination compared to the surface VO$_x$ site.

In situ MES-Raman spectra were also collected to further investigate the structure of the responsive surface metal oxide sites of the catalyst (Fig. S25). Phase-sensitive detection (PSD) of the PL catalyst revealed a strong correlation between the $\nu$(V=O) and $\nu$(W=O) vibrational modes of the VO$_x$ and WO$_x$ surface sites and the introduction of

NH$_3$ (Fig. 4e)[54]. Moreover, it was found that the two bands at 902 cm$^{-1}$ and 930 cm$^{-1}$ were indeed related to bridging V–O–Ti vibrations from two inequivalent adsorption sites (terminal Ti–O and Ti–(OH)–Ti), and not to W–O–Ti vibrations, which was confirmed by conducting similar experiments with the VO$_x$/TiO$_2$ (PL) catalyst (Fig. S26)[71–73]. The terminal V=O phase angle peak (80°) lagged behind that of bridging V–O–Ti vibrations (60°), indicating NH$_3$ activation on V=O would be the RDS in agreement with IR-MES (Fig. S27). Furthermore, the IR phase angle of nitrate (Fig. S24) being out-of-phase with the diminishment of V–O–Ti suggests NH$_3$ adsorption perturbs bridging V–O–Ti and not NO. With aid of theoretical calculations, it was found that the adsorption of nitrate on dimer VO$_x$ resulted in the charge redistribution and decrease in the covalence of V–O bonds, which caused the formation of weak IR bands from the bridging V–O–Ti vibrations (Fig. S28).

Upon integrating the findings from in situ spectroscopic studies, the introduction of NO was observed to prompt the formation of both bridging and bidentate nitrates, as illustrated in Fig. 4a, b. Intriguingly, the adsorption center coincides precisely with the dimeric V sites, a phenomenon distinctly highlighted in Raman spectroscopy by the attenuation of the V=O signal, as shown in Fig. 4e. This particular adsorption pattern highlights the indispensable role of dimeric V sites within PL catalysts, where each V site collaboratively participates in the formation of the nitrate intermediate. Following this, NH$_3$ adsorption leads to the generation of NH$_2$NO species, as captured in the MES-DRIFTS data presented in Fig. 4a. This adsorption event is coupled with the simultaneous formation of bridging hydroxyl groups, revealing the concurrent dehydration process, as delineated in Fig. 4a, b. Literature suggests that the decomposition of NH$_2$NO transpires swiftly[22]. This proposed reaction mechanism accentuates the critical importance of dimeric V sites as the central activation sites for NO adsorption, underlining their significance in the catalytic process.

DFT calculations were conducted to further elucidate the mechanism underlying the superior activity of the PL catalyst. The models of monomeric and dimeric surface vanadia sites on the TiO$_2$(101) anatase surface were constructed and optimized for comparison of the reaction pathway (Fig. S29). The four stages of NH$_3$ and NO-assisted vanadyl reduction (referred to as Red) and one stage of O$_2$ involved in re-oxidation of the reduced surface vanadyl site (referred to as O$_x$), constituting the NO/NH$_3$-SCR reaction cycle is illustrated in Fig. 5. We compared the relative energy profiles of the nitrate-first generation pathway and the NH$_2$-first generation pathway, revealing that the relative energy for NH$_3$ cleavage to form NH$_2$ is higher than that for the nitrate pathway (Fig. S30). Therefore, on dimeric surface vanadia sites, the reaction is more inclined to proceed via the nitrate pathway. With the introduction of NO, as observed by MES-DRIFTS, nitrate species and the disappearance of V=O and V–O–Ti species detected by Raman spectroscopy were noted. This suggests that NO was adsorbed on distinct sites (i.e., bridging V=O with V=O (A→B), V–OH (D→E), or V–OOH (I→J)) and coordinating with bidentate V–O$_2$ (L→M) together with NH$_3$ adsorption. DFT calculations indicate that NH$_3$ has a higher affinity for adsorption on the catalyst surface (Fig. S31), forming lower energy structures, as illustrated by the B configuration in Fig. 5a. Additionally, NH$_2$NO species was observed on the MES-DRIFTS, corresponding to the assistance of surface nitrates for NH$_3$ dehydrogenation into the surface nitroso intermediate (B→C, E→F, J→K and M→N), and subsequently decomposition into N$_2$ and H$_2$O (C→D, F→G, K→L and N→O). MES-DRIFTS observations also revealed M–(OH)–M and M–(OD)–M, corresponding to steps (C→D, F→G, K→L, N→O) involving the dehydration process forming V–OH. Furthermore, adsorbed H$_2$O and D$_2$O were observed, corresponding to step (C→D, O→A, F→G, K→L, N→O) involving the dehydration to form H$_2$O. Additionally, two sub-reaction pathways were involved in the O$_x$ stage: (I) H$_2$O desorption from a reduced vanadia (V*) site (G→H), and (II) gas-phase O$_2$ replenishment on the reduced vanadia (H→I) site. Finally, after the H$_2$O desorbed from V=O groups (O→A), the catalytic cycle was completed. The calculated formation energy of nitrate species on monomeric and dimeric surface vanadia sites are −0.34 eV and −0.47 eV (Fig. S32), respectively. Consequently, comparatively, nitrate

formation is less favorable on monomeric vanadia sites. Additionally, MES-DRIFTS did not detect any nitrate species. Therefore, the monomeric surface vanadia site followed a reaction pathway without nitrate species as shown in Fig. 5b. The rate-determining step for the monomeric surface vanadia site was the V–OOH dehydration (G→H) with 1.44 eV in the Red 2 stage, which is in agreement with previous reported monomeric vanadia/TiO$_2$ surfaces[22]. However, the rate-determining step changed to the transition state of NH$_3$ dehydrogenation on the V=O bond with V–OH (E→F) with 1.17 eV over dimeric surface vanadia sites, which was also in the stage of vanadyl reduction. To further investigate the dehydration behavior for dimeric surface vanadia, the transition states for the generation of H$_2$O (G→H and N→O processes) are calculated in Fig. S33. It is shown that their energy barriers for G→H (TS5) and N→O (TS6) are 0.98 eV and 1.02 eV, respectively, which is indeed lower than the relative energy of 1.17 eV for TS2 (Figs. 5 and S33.) The results indicated that dimeric surface vanadia also significantly reduced the energy barrier for H$_2$O desorption.

In situ UV−Vis time-resolved spectroscopy was further employed to compare the V=O reduction step for PL and OR catalysts[19,74]. The kinetics of V$^{5+}$ reduction was determined by monitoring the percentage of reduced V$^{5+}$ sites under NH$_3$ and NH$_3$−NO exposure conditions at 150 °C, using the *d-d* transition band of reduced vanadia at 799 nm as a reference (Figs. S34, S35, and Table S6). It was observed that the specific reduction rates of V$^{5+}$ for the PL catalyst ($0.76 \times 10^{-2}$ min$^{-1}$) were approximately twice as high as those for the OR catalyst ($0.40 \times 10^{-2}$ min$^{-1}$). This finding further suggests that the improved kinetics of the surface V$^{5+}$ reduction step is highly correlated with the enhanced catalytic activity of the PL catalyst.

To uncover the factors contributing to the decline in activity following prolonged plasma treatment, we analyzed the spectral characteristics and reducibility of the PL-300s catalyst. Raman spectra (Fig. S36a) revealed a significant decrease in terminal V=O bonds in the PL-300s sample. V 2$p$ XPS analysis (Fig. S36b) indicated a shift in the valence state of V from 5+ to 4+. H$_2$-TPR profiles (Fig. S36c) showed a reduction in hydrogen consumption for the V reduction peak, suggesting a suppression of V's redox ability from 1.16 cm$^3$/g in PL to

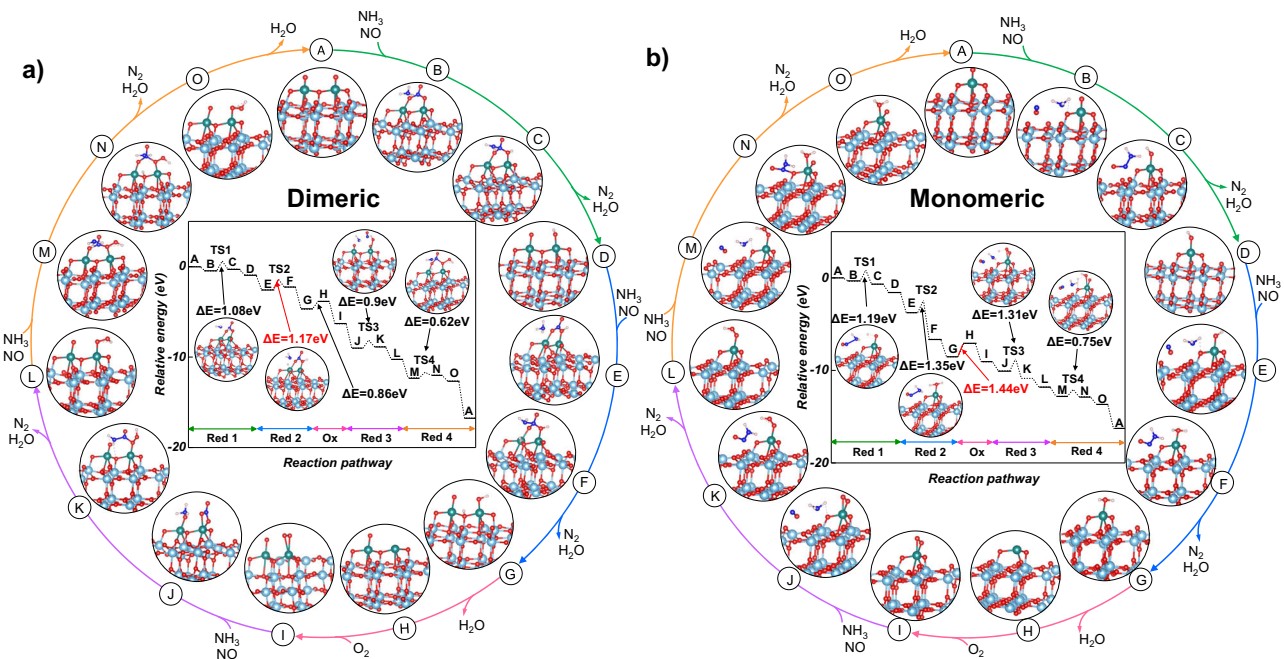

**Fig. 5 | DFT calculations for the reaction pathway of SCR over dimeric and monomeric vanadia.** The optimized molecular structures for the reactant, transition states, intermediates, product, and reaction energies were determined using DFT theory for each elementary step in the NH$_3$-SCR mechanism over the surfaces of (**a**) dimeric surface vanadia site and (**b**) monomeric surface vanadia site. Red, cyan, green, blue, and white circles denote O, Ti, V, N, and H atoms, respectively.

0.63 cm$^3$/g in PL-300s. In-situ UV–Vis experiments (Fig. S36d) demonstrated a decrease in reducible V$^{5+}$ content to 11.8% in PL-300s after introducing NH$_3$ and NO, indicating fewer active V sites. The deterioration of the NH$_3$-SCR activity of the catalyst is exacerbated by the prolonged plasma treatment, as the destruction of V=O is pivotal for activating NH$_3$ and promoting the generation of NH$_2$NO (steps B→C, E→F, J→K, M→N)[19,40,75].

In summary, supported vanadia nanocluster catalysts were successfully fabricated by transformation from monomeric surface VO$_x$ sites from a classic supported V$_2$O$_5$–WO$_3$/TiO$_2$ catalyst via a H$_2$ plasma treatment. The plasma treatment resulted in the generation of a distorted lattice shell overlayer on the surface of the TiO$_2$ support. This facilitated surface migration and reconstruction of surface VO$_x$ and WO$_x$ sites on the titania support under an H$_2$-reducing atmosphere. The atomic-scale distribution of oligomeric surface vanadia sites on TiO$_2$ were identified by the combination of HAADF-STEM and EELS microscopy. The plasma-treated supported oligomeric surface vanadia sites exhibited superior SCR activity, selectivity, and long-term stability in comparison to the conventional supported SCR VWTi catalyst. Moreover, the novel plasma-assisted method also significantly enhanced the activity of other supported vanadia-based catalysts. The structural investigations indicate that the oligomerization of the surface vanadyl sites was not caused by just the plasma treatment of the TiO$_2$ support, but required the coexistence of vanadia and titania in the plasma reduction atmosphere to trigger aggregation of monomeric surface VO$_x$ sites at the TiO$_2$ surface defects to generate strong interactions between the active components and oxide support. The MES investigations revealed that oligomeric surface VO$_x$ sites provide exclusive centers for adsorbed bridging and bidentate nitrates, and assisted in NH$_3$ activation to generate amide intermediates. DFT calculations revealed that the enhanced activity from oligomerization of surface vanadyl sites is related to the barrier-less steps of the V$^{5+}$ reduction from NH$_3$ dehydrogenation with the assistance of adsorbed nitrates. This research contributes to a deeper understanding of structure-activity relationship and reaction mechanism of the widely used supported vanadia-based/TiO$_2$ catalysts for NH$_3$-SCR.

# Methods

## Catalyst preparation
Tungsten and vanadium were deposited on anatase-TiO$_2$ (4.75 g, >99.8% anatase, Maklin Co.) using tungsten oxalate (0.2325 g, Maklin Co.) and ammonium metavanadate (0.0645 g, Maklin Co.) dissolved in an aqueous oxalic acid solution (1 mol/l). Typically, water was removed slowly by using a rotary evaporator, and the obtained solid was dried at 100 °C overnight and calcined at 500 °C for 3 h in air. Plasma treatment was performed in a PECVD system (KeJing co. Anhui, China) equipped with a high-frequency generator operating at 13.56 MHz and a power of up to 500 W. 100 mg of the OR catalyst powder was evenly spread on the quartz plate and introduced into the plasma chamber for x s, the H$_2$/Ar/O$_2$ was introduced as a pulse into the chamber. H$_2$ was slowly introduced into the chamber before plasma treatment until atmospheric pressure was reached. Before commencing the plasma treatment procedure, the chamber pressure was reduced to 10 Pa employing a vacuum pump. Subsequently, the power supply was incrementally raised to 500 w, employing a slow and controlled approach. At this stage, the timing protocol was initiated. After 10-s plasma treatment, hydrogen gas was reintroduced into the chamber. The aforementioned steps were iterated multiple times to attain the intended duration for modification. Subsequently, the samples were calcined at 500 °C for 3 h in air.

## Catalyst characterization
HAADF images of the samples were obtained using a ThermoFisher Themis Z transmission electron microscope with a convergence angle of 25 mrad and inner and outer collection angles of 59 and 200 mrad,

respectively. To acquire the spectroscopic data needed for EELS elemental mapping, the electron probe (in our setups the probe has a diameter of ~1.0 Å) was scanned in cluster regions and an EELS spectrum (350–850 eV) was acquired at each point together with HAADF image as reference. After using the average spectra as individual components in a linear combination, the spectra were fitted, and 2D atomic maps of the spectral weights were generated in combination with the simultaneously acquired HAADF image. XPS was performed with an XPS spectrometer (Thermo, Escalab 250Xi, USA) with Al Kα radiation. The temperature-programmed reduction with H$_2$ (H$_2$-TPR) experiments was carried out on a chemisorption instrument. Before conducting the testing, the catalyst samples were subjected to a pre-treatment at 300 °C for 60 min using helium as carrier gas. This pre-treatment was performed to remove moisture and impurities from the samples. (Micromeritics, AutoChem II 2920, USA). Quasi in situ HS-LEIS spectra were obtained using the Qtac$^{100}$ HS-LEIS spectrometer (ION-TOF) equipped with a highly sensitive double toroidal analyzer. Using a Bruker Avance III 500 spectrometer with a resonance frequency of 131.6 MHz, the $^{51}$V solid-state NMR tests were performed at 11.7 T. A 1.9-mm HX double-resonance probe was utilized at a spinning rate of 40 kHz. The in situ $^{51}$V NMR spectra of the dehydrated samples were carried out employing a Hahn−echo pulse sequence, with a π/2 pulse width of 1.5 μs. For the present samples, a total of 60,000 scans were conducted, with a recycle delay of 0.3 s between each scan.

## Measurement of NH$_3$-SCR activity and kinetics
The NH$_3$-SCR activity and kinetic data were measured with a tubular quartz reactor system, TOF are calculated by dividing the amount of NO molecules converted per second at low NO conversion (<15%) by the per V atoms on the surface of catalysts. (Additional details are provided in the supplementary information Section 2)[60]. Outlet NO, NO$_2$, NH$_3$, N$_2$O, SO$_2$, and H$_2$O concentrations were monitored by a Fourier-transformed infrared spectrometer (MBGAS-3000; ABB Co.)[75].

## MES (modulation excitation spectroscopy) experiments
In situ DRIFTS was performed using a FT-IR spectrometer (Thermo Fisher Scientific, Nicolet 6700) equipped with a mercury−cadmium telluride detector and a low void volume cell (Jiaxing Puxiang Tech. Ltd, RC-DRS -K01). The thermocouple was directly placed into the catalyst powder for temperature measurement. For the concentration modulation excitation experiments, the solenoid valves were used to automatically switch between gases. The pulse sequence according to Fig. S17 (NO + O$_2$/NH$_3$ + O$_2$ modulation: 2000 ppm NO/Ar *vs.* 2000 ppm NH$_3$/Ar, constant 5% O$_2$/Ar) was introduced into the reaction cell. The set of time-resolved spectra obtained from the modulation experiments was converted into MES spectra using PSD:

$$I(\phi^{PSD})n_v = \frac{2}{T}\int_0^t I(t)\sin(k\omega t + \phi^{PSD})dt \qquad (1)$$

where $I$(t) is the set of time-resolved data, $\omega$ the stimulation frequency, $k$ the demodulation index ($k$ = 1 is the fundamental harmonic and was used in this work), $T$ the modulation period, and $\phi^{PSD}$ the phase angle. Python was used to process the time-resolved data using PSD. The modulation period ($T$ = 240 s) is defined as the time required to conclude one full sequence. A single modulation period typically consisted of 240 consecutive time-resolved FTIR spectra, identical modulation sequences were applied and consisted of 12 consecutive. The FTIR spectra were recorded as 8 scans at a resolution of 4 cm$^{-1}$ [62,63,76].

## In situ Raman spectroscopic characterization
Raman spectra were carried on a Senterra II Raman spectrometer (Bruker Optic), with an excitation wavelength of 532 nm and a low void volume cell (Jiaxing Puxiang Tech. Ltd, RC-RAMAN-K01). For the

time-resolved experiments, the sample was first treated in a 10% $O_2$/Ar (50 ml/min) flow at 500 °C for 30 min. Then the reaction cell was cooled to 150 °C (200 °C) in an Ar flow (50 ml/min) and collected as background spectra. Then the catalyst was sequentially exposed to 5% $O_2$ (50 ml/min), 2000 ppm of $NH_3$ (50 ml/min), 2000 ppm of $NH_3$ + 2000 ppm of NO (50 ml/min), and 2000 ppm of $NH_3$ + 2000 ppm of NO (50 ml/min) + 5% $O_2$ (50 ml/min). The Raman spectra were recorded every 2 min at a resolution of 2 cm$^{-1}$. The same pulse sequence and data processing methods as FTIR were employed for the Raman concentration modulation excitation experiments. The Raman spectra were recorded every 30 s at a resolution of 4 cm$^{-1}$.

## Computational details

First-principles calculations were performed using the DFT framework within the Vienna ab initio simulation package (VASP 5.4.4)[77–79]. A (3 × 1) supercell of the anatase (101) surface with double layer was employed as substrate for the commercial SCR catalyst surface[80–82]. The thickness of vacuum layer of anatase (101) surface was set over 15 Å. For relaxation of vanadia-loaded anatase surface and gases absorbed models, atoms at bottom eight layers were fixed, which means the upper vanadia clusters were allowed to relax and interact with gas molecules. After geometric optimization with generalized gradient approximation Perdew–Burke–Ernzerhof (GGA-PBE) functionals, the lattice parameters became 11.28 Å × 9.94 Å × 20.31 Å, which is in good agreement with input experimental lattice parameters (11.33 Å × 10.2 Å × 20.84 Å) of anatase (101) surface shown in Fig. S29. PBE functionals, based on the GGA, were widely used to account for exchange-correlation of $V_2O_5$/$TiO_2$ catalyst for selective catalytic reduction with ammonia[22,82–85]. The interaction between the ions and the electrons was described by projector-augmented wave methods[85]. The pseudopotentials used for the present models were constructed by the electron configurations as V $3s^2sp^63d^44s^1$ states, Ti $3s^2sp^63d^24s^2$ states, N $2s^22p^3$ states, H $1s$ states, and O $2s^22p^4$ states. The energy cut-off value was set at 600 eV[86]. The convergence criteria of total energies and forces were $10^{-6}$ eV/atom and 0.05 eV/Å. The first Brillouin zone was sampled by a Monkhorst–Pack 2 × 2 × 1 K-point mesh[87]. The adsorption energies and electron density difference were calculated according to the adsorption or interfacial models[88–91]. We used dimeric vanadyl species as the model for our DFT calculations because they are the basic structural unit of various polymeric vanadia structures and can reasonably represent the coupling effect in them. The coupling effect between two adjacent vanadyl species (i.e., within a dimer unit of vanadia) at the reaction site was common in dimeric and higher-order polymeric vanadia structures. It sped up the whole catalytic cycle during the $NH_3$-SCR of NO over the polymeric vanadyl species, and thus, we expected that dimeric and higher-order polymeric vanadia would have similar effects on the SCR reaction. Free energy correction was performed by including the zero-point energy and enthalpic and entropic contributions from vibrational degrees of freedom, with the substrate fixed. Climbing Image Nudged Elastic Band method was employed to find the minimum energy path connecting the reactants and products[91–93]. The fast inertial relaxation engine was used as optimizer in CI-NEB.

## Data availability

The data generated within the paper and its Supplementary Information file are available from the corresponding authors upon request. Source data of Figs. 2–4 are provided in a Source Data file. Source data are provided with this paper.

## Code availability

All code used in the simulations supporting this paper is available from the respective authors upon request.

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

## Acknowledgements

This work was financially supported by the National Natural Science Foundation of China (22176008, 21906004, and 52202154). Bingcheng Luo acknowledges support from the High-performance Computing Platform of China Agricultural University. We thank Si Jiang for the scientific discussion and valuable suggestions. We thank Didi Li at East China University of Science and Technology for their help with X-ray absorption spectroscopy characterization. We thank Yuan Xu (Bruker Co.) for assistance with collecting Raman spectra. We acknowledge support from National Supercomputer Center in Tianjin, and the energy calculations were performed on Tianhe new generation supercomputer. We thank Xiumei Wang (Bruker NMR Facility) for assistance with collecting NMR spectra. The work at Lehigh University was supported as part of Understanding & Control of Acid Gas-Induced Evolution of Materials for Energy (UNCAGE-ME), an Energy Frontier Research Center funded by the U.S. Department of Energy, Office of Science, Basic Energy Sciences under Award # DE-SC0012577.

## Author contributions

X.L. and Y.Y. proposed and designed the research plan and experimental scheme. Y.Y., K.L., M.Z., I.E.W. and X.L. performed the characterization of the materials. Y.Y. carried out the synthesis and the activity tests of the materials and data analysis. Y.Y., X.L., K.L., M.Z., B.M.L., Y.S. and I.E.W. conducted mechanistic investigation experiments and analysis. B.L. performed the first-principles calculations. X.L., M.Z., I.E.W., B.L., K.L., T.Z. and Y.Y. co-wrote the manuscript. All authors discussed the results and provided comments on the manuscript.

## Competing interests

The authors declare no competing interests.
