## [Peer Review File · Nature Communications]

Plasma-Assisted Manipulation of Vanadia Nanoclusters for Efficient Selective Catalytic Reduction of NO_xREVIEWER COMMENTS

Reviewer #1 (Remarks to the Author):

The manuscript by Yin et al. on "Plasma-Assisted Manipulation of Vanadia Nanoclusters for Efficient Selective Catalytic Reduction of NO_x" presents a novel plasma-assisted method for synthesizing supported metal oxide nanoclusters, exemplified by transforming V₂O₅-WO₃/TiO₂ catalyst. The resulting nanoclusters, observed through advanced electron microscopy, significantly enhance turnover frequency for selective catalytic reduction (SCR) of NO with NH₃. The work on the topic is interesting, however, some specific details, as given below, are required to further improve the quality of the manuscript.

Comments

- The author mentioned that "the NH₃-SCR reaction pathway was found for the first time to involve surface nitrate reaction intermediates" However, such a study has been performed before and nitrate species have been detected earlier as reported in these articles. ACS Catal. 2015, 5, 5, 2832–2845, Chem. Sci., 2020, 11, 447-455
- How do the spectroscopically identified species support the mechanism?
- In the section discussing the use of isotopically labelled reactant ND₃, can the author elaborate on how this method contributes to the avoidance of overlap from NH₃ adsorption bands?
- Did the author benchmark the PBE functional against the experimental? Furthermore, did the author conduct any comparisons with hybrid functionals, for example, PBE vs PBE0, to assess their performance in describing the system?
- Has the computational study been validated against experimental data, specifically considering the proposed NH₃-SCR reaction, the overall experimental $\Delta_f H^\circ$ value calculated from the standard enthalpy of formation of the gaseous species involved in the reaction can be compared with the calculated reaction energies?
- The adsorption or binding energies for the gaseous molecules with surfaces are missing- Further, comparing these values with the reported experimental heat of adsorption values for the corresponding species will strengthen the link between theoretical values and experimental data.
- Overall, the mechanism is quite unclear- the author mentioned that "the monomeric surface vanadia site followed a similar reaction pathway except that adsorbed NH₃ interacted with gaseous NO as compared to the dimeric surface. On what basis author conclude this? No clear explanation is given; for example, what are the formation energies of important intermediate species such as nitrate?
- Given the intricacy of SCR reactions, can you elaborate on plans for more extensive mechanistic studies to enhance understanding, especially regarding the suggested N-H cleavage and its consequences on surface species? For instance, is there consideration for exploring the potential formation of nitrosamine (N(=O)-NH₂) species as a result of the proposed N-H cleavage?
- Author should be careful about miss typos as can be seen in line (74) "However, as a case of For the commercial..". Also, some missing ref, for example, in line (250) "In contrast to Marberger et al.'s study, isotope experiment confirmed that the band at 1599 cm⁻¹ should be attributed to nitrate rather than NH₃, suggesting the involvement of nitrate in the reaction". Also, I don't see the information in Figure S26 (SI information) as stated by the author in this statement "The lattice dimension was about 11.28Å × 9.94Å × 20.31Å. V/W oxide clusters were loaded on the substrate to form a VWTi catalyst as

shown in Figure S26". Further details need to be added about the chosen modeled systems.

Reviewer #2 (Remarks to the Author):

This study presents a plasma-assisted treatment strategy to promote the oligomerization of vanadium species over VWTi catalysts. The activity enhancement mechanism of the oligomeric vanadia-dominated catalyst was characterized through various experimental measurements and DFT calculation. This topic is interesting and important. However, the manuscript must address the following questions before publication in Nature Communications.

Specific comments are listed below :

- (1) The plasma treatment resulted in the polymerization of V species. But the reason and the mechanism were not clarified in this manuscript.
- (2) Line 174-175: With longer treatment time, the reaction rate decreased. Why? Are there any changes in the vanadyl species with the change of treatment time? The authors did not provide an explanation or discussion.
- (3) In Figure 2, PL presented higher ratio of polymeric V species and lower H₂ reduction temperature. It is contradictory. In general, the monomeric V can be reduced by H₂ at lower temperatures.
- (4) In Line 138, the PL catalyst also exhibited a broad band centered at 930 cm⁻¹ assigned to the V-O vibration from the bridging V-O-Ti bond. However, it is different from the Figure 2, which show V-O-V structure.
- (5) In Figure 2d, the fitting of NMR was unreasonable for the noticeably different FWHM between the two catalysts.
- (6) Line 231: The peak at 1330 cm⁻¹ was attributed to NH₂NO species, which is inconsistent with the literature. Previous studies have found that the characteristic peak of NH₂NO is at 1490 cm⁻¹ over VWTi catalysts (Angew. Chem., Int. Ed. 2016, 55, 11989).
- (7) Lines 266-270: L-NH₃ (i.e., NH₃ adsorbed on surface metal sites) is considered as the primary active species involved in the SCR reaction, which is inconsistent with the adsorption sites of NH₃ in the DFT calculation models.
- (8) Line 328: The dehydration process is the rate-determining step, which is unbelievable. If the effect of entropy is considered (i.e., calculating the free energy), the desorption energy of water molecules would be much smaller.
- (9) DFT calculation method section: The thickness of the vacuum layer, whether the bottom layer of the model is fixed, and the calculation method for the transition state are not defined.
- (10) Figure 5a: The transition states for the generation of H₂O (G→H and N→O processes) have not been provided. In fact, the two -OH groups are so far apart that H cannot directly transfer to form one water molecule.

Reviewer #3 (Remarks to the Author):

The authors successfully demonstrated the preparation of V₂O₅-WO₃/TiO₂ by plasma treatment and its enhanced activity in NH₃-SCR. Many characterization results have revealed the state of V₂O₅-WO₃/TiO₂, and detailed studies have been conducted. However, some improvements are needed to facilitate reader understanding. Detailed comments are given below.

1. In the same sentence on line 74, 'however' is used twice.
2. Fig. 1a is not mentioned in the text. Please add it to the text.
3. Fig. S6 is PL for both.
4. Please explain which color is the surface side in Fig. S7 and S8.

5. In line 170, the value of GHSV is different from SI. Also, please provide the actual flow rate, since some readers may refer to the actual flow rate.
6. In line 200, indicate which sample was calcined.
7. In lines 202-207, a control experiment is performed, but the sample names are abbreviated in the SI. Please explain which sample corresponds to which sample name.
8. The term 'carrier' is used in line 212, whereas 'support' is usually used in this field.

Point-to-point response to Editors and Reviewers

Reviewer: 1

Comments:

The manuscript by Yin et al. on “Plasma-Assisted Manipulation of Vanadia Nanoclusters for Efficient Selective Catalytic Reduction of NO_x” presents a novel plasma-assisted method for synthesizing supported metal oxide nanoclusters, exemplified by transforming V₂O₅-WO₃/TiO₂ catalyst. The resulting nanoclusters, observed through advanced electron microscopy, significantly enhance turnover frequency for selective catalytic reduction (SCR) of NO with NH₃. The work on the topic is interesting, however, some specific details, as given below, are required to further improve the quality of the manuscript.

Comment 1-1. The author mentioned that “the NH₃-SCR reaction pathway was found for the first time to involve surface nitrate reaction intermediates” However, such a study has been performed before and nitrate species have been detected earlier as reported in these articles. ACS Catal. 2015, 5, 5, 2832–2845, Chem. Sci., 2020, 11, 447-455

Response 1-1:

Thank you for bringing this to our attention, and we appreciate the valuable feedback. We acknowledge that our statement regarding the "NH₃-SCR reaction pathway being found for the first time to involve surface nitrate reaction intermediates" may have overlooked previous relevant studies.

Ton V. W. Janssens et al. discuss the role of a nitrate/nitrite equilibrium and the potential influence of Cu dimers and Brønsted sites in their research.^[1] Greenaway A G and colleagues' research on Cu-SSZ-13, employing modulation excitation (ME) DRIFTS and XANES techniques, reveals key intermediates ([Cu²⁺(OH-)]⁺, Cu-N(=O)-NH₂, Cu-NO₃) in the NH₃-SCR process, offering mechanistic insight. These intermediates are pivotal in the standard SCR cycle, contributing to the production of the desired product (N₂) and demonstrating Cu ion reoxidation.^[2] Negri C. et al. identified the bands of the monodentate nitrate group linked to Cu ions, while confirming their strong interaction with NH₃ molecules.^[3] Yao L et al. revealed that the

formation of bridging nitrates could also accelerate the reaction with adsorbed NH_3 species and be favorable for further low temperature NH_3 -SCR reaction. [4] The addition of Ce into the VWTi catalyst, as observed by Chen L. et al., led to an enhancement of the basic property on the TiO_2 support, resulting in an increased availability of ad- NO_x species for the SCR reaction. [5] Liu Z. et al. found that the existence of Ce on the V0.5Ce5Ti catalyst can promote the formation of NO_2 and monodentate nitrate species, both of which were reactive intermediates for the NH_3 -SCR of NO_x . [6] Cheng J et al. discovered that the microwave-treated V_2O_5 @AC-300 catalyst possesses superior redox ability, which is favorable for the $\text{V}^{5+} \leftrightarrow \text{V}^{4+}$ cycle. Additionally, the $\text{V}^{4+}/(\text{V}^{4+} + \text{V}^{5+})$ proportion was increased by adjusting the interaction between vanadium and oxygen atoms with the aid of microwaves, leading to more chemisorbed oxygen and frequent oxygen migration, which heightens the activation of reactants and NO oxidation to NO_2 and nitrates. [7]

Despite extensive research recognizing the crucial role of nitrate species in the NH_3 -SCR reaction, current studies have not explained, at the atomic level, how nitrate species participate in the NH_3 -SCR reaction or established the relationship between nitrate generation and catalyst structure. Therefore, utilizing modulation excitation spectroscopy (MES) technologies along with density functional theory (DFT) theoretical calculations, we identified the generation of nitrate species on dimeric V sites. Furthermore, we demonstrated their further activation to form NH_2NO species, thereby promoting low-temperature NH_3 -SCR reactions.

We have carefully reviewed and cited the references mentioned above, and the expression "for the first time" has been removed from **line 97** of the manuscript. Additionally, relevant descriptions have been added in **lines 80-86**.

References:

1. Janssens T V W, Falsig H, Lundegaard L F, et al. A consistent reaction scheme for the selective catalytic reduction of nitrogen oxides with ammonia. *ACS catalysis*, 2015, 5(5): 2832-2845.
2. Greenaway A G, Marberger A, Thetford A, et al. Detection of key transient Cu intermediates in SSZ-13 during NH_3 -SCR de NO_x by modulation excitation IR

spectroscopy. *Chemical Science*, 2020, 11(2): 447-455.

3. Negri C, Borfecchia E, Cutini M, et al. Evidence of Mixed - Ligand Complexes in Cu-CHA by Reaction of Cu Nitrates with NO/NH₃ at Low Temperature. *ChemCatChem*, 2019, 11(16): 3828-3838.

4. Yao L, Liu Q, Mossin S, et al. Promotional effects of nitrogen doping on catalytic performance over manganese-containing semi-coke catalysts for the NH₃-SCR at low temperatures. *J. Hazard. Mater.*, 2020, 387: 121704.

5. Chen L, Li J, Ge M. Promotional effect of Ce-doped V₂O₅-WO₃/TiO₂ with low vanadium loadings for selective catalytic reduction of NO_x by NH₃. *The Journal of Physical Chemistry C*, 2009, 113(50): 21177-21184.

6. Liu Z, Zhang S, Li J, et al. Novel V₂O₅-CeO₂/TiO₂ catalyst with low vanadium loading for the selective catalytic reduction of NO_x by NH₃. *Applied Catalysis B: Environmental*, 2014, 158: 11-19.

7. Cheng J, Xu R, Song L, et al. Unveiling the role of microwave induction on V₂O₅@AC catalysts with enhanced activity for low temperature NH₃-SCR reaction: an experimental and DFT study. *Environmental Science: Nano*, 2023, 10(5): 1313-1328.

Comment 1-2 How do the spectroscopically identified species support the mechanism?

Response 1-2:

We appreciate the reviewer's inquiry regarding the support for the proposed mechanism through spectroscopically identified species. In Fig. 4a and 4b, upon the introduction of NO, bridging nitrates (1288 cm^{-1} vs(N-O)₂ and 1599 cm^{-1} vas(N-O)₂) and bidentate nitrates (1260 cm^{-1} vs(N-O)₂ and 1579 cm^{-1} vas(N-O)₂) were observed. Simultaneously, in Fig. 4e, Raman peaks of V=O (1030 cm^{-1}) and V-O-Ti (902 and 930 cm^{-1}) disappeared, corresponding to steps (A→B, D→E, I→J, L→M) where NO adsorbs on dimeric V sites to form nitrates. Additionally, NH₂NO species (1330 vs(N-H) and 1490 cm^{-1} v(N=O)), observed on the MES-DRIFTS, correspond to steps (B→C, E→F, J→K, M→N), where NH₃ reacts with nitrates to generate NH₂NO. MES-DRIFTS observations also revealed M-(OH)-M (3653 cm^{-1}) and M-(OD)-M (2690 cm^{-1}), corresponding to steps (C→D, F→G, K→L, N→O) involving the dehydration process forming V-OH. Furthermore, adsorbed H₂O and D₂O were observed, corresponding to step (C→D, O→A, F→G, K→L, N→O) involving the dehydration to form H₂O.

By summing up the *in situ* spectroscopic results, upon the introduction of NO, bridging nitrates and bidentate nitrates were observed as in Fig. 4a and 4b, where the adsorption center is exactly the center of dimeric V sites as disclosed in Raman where V=O signal dampens as in Fig. 4e. This adsorption mode suggests the critical role of the dimeric V sites in PL catalysts, where both V sites are involved to form nitrate intermediate. Sequentially, NH₃ adsorbs to form NH₂NO species as observed on the MES-DRIFTS in Fig. 4a, while simultaneously forming bridging hydroxyl as is proved in Fig. 4a&b, disclosing the dehydration process along side with NH₃ adsorption. The decomposition of NH₂NO is a fast process as suggested by literatures [8-10]. The suggested reaction mechanism stresses the necessity of dimeric V sites as the activation center for NO adsorption. The relevant statement has been added to **lines 334-344 and lines 351-368** in the manuscript.

References:

8. Soyer S, Uzun A, Senkan S, et al. A quantum chemical study of nitric oxide reduction

by ammonia (SCR reaction) on V_2O_5 catalyst surface. *Catal. today*, 2006, 118(3-4): 268-278.

9. Mason M M, Lee Z R, Vasiliu M, et al. Initial steps in the selective catalytic reduction of NO with NH_3 by TiO_2 -supported vanadium oxides. *ACS Catal.*, 2020, 10(23): 13918-13931.

10. Chen W, Xu S. Unraveling the Complete Mechanism of the NH_3 -Selective Catalytic Reduction of NO over CeO_2 . *ACS Catal.*, 2023, 13: 15481-15492.

Comment 1-3. In the section discussing the use of isotopically labelled reactant ND₃, can the author elaborate on how this method contributes to the avoidance of overlap from NH₃ adsorption bands?

Response 1-3:

Due to overlapping infrared signals around 1600 cm⁻¹ from bidentate nitrate (ν_{as} (N-O)₂ at 1579 cm⁻¹), bridging nitrate (ν_{as} (N-O)₂ at 1599 cm⁻¹), L-NH₃ (1604 cm⁻¹), and H₂O (1618 cm⁻¹), accurate identification is challenging when NH₃ and NO coexist. To address this, MES-DRIFTS studies used isotopically labeled ND₃ to avoid signal overlap with NH₃, nitrate, and H₂O absorption bands.

IR peaks associated with NH₃ shift to lower wavenumbers due to the spectral isotope effect of ND₃. Notably, differences in the isotope nuclear mass result in different vibrational frequencies, causing shifts in the infrared spectrum.^[11] From the harmonic oscillator model, the relationship between vibrational frequencies and mass and force constant is as below:

$$\nu = \frac{1}{2\pi} \sqrt{\frac{K}{M}}$$

K is the force constant

For diatomic molecules (m_1, m_2), $M = \frac{m_1 m_2}{m_1 + m_2}$

$$\nu_{N-H} = \frac{1}{2\pi} \sqrt{\frac{K}{M(N-H)}}$$

$$\nu_{N-D} = \frac{1}{2\pi} \sqrt{\frac{K}{M(N-D)}}$$

$$\frac{\nu_{N-H}}{\nu_{N-D}} = \sqrt{\frac{M(N-D)}{M(N-H)}} = 1.37$$

Marberger *et al.* identified NH₃* at 1603 cm⁻¹ using MES-FTIR.^[12] After employing ND₃ as the reactant, the adsorption peak of ND₃* shifted to 1170 cm⁻¹. Additionally, the position of D₂O shifted from the original H₂O IR peak at 1618 to 1385 cm⁻¹. However, the infrared absorption peak positions of bidentate nitrate and bridging nitrate remained at 1579 and 1599 cm⁻¹. Therefore, by using isotopically labeled reactant ND₃,

the influence of adsorbed NH₃ and H₂O infrared peaks was excluded, enabling the accurate identification of nitrate species. The role of the isotopically labeled reactant ND₃ has been incorporated into **lines 267-270** of the manuscript and in **section 7** in the supporting information.

References:

11. Koops, Th. *et al.* Measurement and Interpretation of the Absolute Infrared Intensities of NH₃ and ND₃. *J. Mol. Struct.* 1983, 96: 203-218.
12. Marberger, A. *et al.* The significance of lewis acid sites for the selective catalytic reduction of nitric oxide on vanadium-based catalysts. *Angew. Chem. Int. Ed. Engl.* 2016, 55:11989-11994.

Comment 1-4. Did the author benchmark the PBE functional against the experimental? Furthermore, did the author conduct any comparisons with hybrid functionals, for example, PBE vs PBE0, to assess their performance in describing the system?

Response 1-4:

Thanks for your suggestion.

A (3x1) supercell of the anatase (101) surface with double layer was employed as substrate for the commercial SCR catalyst surface. ^[13, 14] The thickness of vacuum layer of anatase (101) surface was set over 15Å. After geometric optimization with GGA-PBE functionals, the lattice parameters became 11.28Å × 9.94Å × 20.31Å, which is in good agreement with input experimental lattice parameters (11.33Å × 10.2Å × 20.84 Å) of anatase (101) surface. ^[15] Hybrid functional such as PBE0 and HSE06 was not employed, because the lattice parameters of anatase is systematically overestimated with most quantum chemical methods as shown by Esch et al and Labat et al. ^[16,17]

Moreover, Perdew–Burke–Ernzerhof (PBE) functionals, based on the generalized gradient approximation (GGA), were widely used to account for exchange–correlation of V2O5/TiO2 catalyst for selective catalytic reduction with ammonia. ^[18,19]

References:

13. Mino L, Cazzaniga M, Moriggi F, et al. Elucidating NO_x Surface Chemistry at the Anatase (101) Surface in TiO₂ Nanoparticles. *The Journal of Physical Chemistry C*, 2022, 127(1): 437-449.
14. Langhammer D, Kullgren J, Österlund L. Photoinduced adsorption and oxidation of SO₂ on anatase TiO₂ (101). *Journal of the American Chemical Society*, 2020, 142(52): 21767-21774.
15. Weirich T E, Winterer M, Seifried S, et al. Rietveld analysis of electron powder diffraction data from nanocrystalline anatase, TiO₂. *Ultramicroscopy*, 2000, 81(3-4): 263-270.
16. Esch T R, Gadaczek I, Bredow T. Surface structures and thermodynamics of low-index of rutile, brookite and anatase–A comparative DFT study. *Applied Surface*

Science, 2014, 288: 275-287.

17. Labat F, Baranek P, Domain C, et al. Density functional theory analysis of the structural and electronic properties of TiO₂ rutile and anatase polytypes: Performances of different exchange-correlation functionals. *The Journal of chemical physics*, 2007, 126(15).

18. He G, Lian Z, Yu Y, et al. Polymeric vanadyl species determine the low-temperature activity of V-based catalysts for the SCR of NO_x with NH₃. *Science advances*, 2018, 4(11): eaau4637.

19. Song I, Lee H, Jeon S W, et al. Simple physical mixing of zeolite prevents sulfur deactivation of vanadia catalysts for NO_x removal. *Nature communications*, 2021, 12(1): 901.

Comment 1-5. Has the computational study been validated against experimental data, specifically considering the proposed NH₃-SCR reaction, the overall experimental $\Delta_f H^\circ$ value calculated from the standard enthalpy of formation of the gaseous species involved in the reaction can be compared with the calculated reaction energies?

Response 1-5:

Thanks for your suggestion.

Owing to the low concentrations of our reactants (NH₃: 350 ppm, NO: 350 ppm), directly measuring the heat released during the catalytic process proves challenging experimentally. Hence, we employ the difference in standard molar enthalpies of formation between reactants and products to calculate the molar enthalpy change of the reaction. The overall reaction equation for the NH₃-SCR catalytic reaction cycle that we have designed is:

Upon consulting the Lange's Handbook of Chemistry, the standard molar enthalpies of formation for the reactants and products are provided in Table 1. ^[20]

reactants/products	$\Delta_f H$ (kJ/mol)
NH ₃	-45.94
NO	91.29
O ₂	0
N ₂	0
H ₂ O	-241.8

Hence, the overall molar enthalpy change for the reaction is:

$$\Delta H = (-241.8) \times 6 - 91.29 \times 4 - (-45.94) \times 4 = -1632.2 \text{ kJ/mol}$$

The total molar enthalpy change obtained through DFT calculations for the reaction is -1700.9 kJ/mol within the consideration of thermodynamic zero-point energy (ZPE) corrections and entropy contributions, which closely approximates the molar reaction enthalpy calculated through consulting the Lange's Handbook of Chemistry.

References:

20. Speight J G. Lange's handbook of chemistry. New York: McGraw-hill, 2005.

Comment 1-6. The adsorption or binding energies for the gaseous molecules with surfaces are missing- Further, comparing these values with the reported experimental heat of adsorption values for the corresponding species will strengthen the link between theoretical values and experimental data.

Response 1-6:

Thanks for your suggestion.

The absorption energy of the upper gaseous molecules is shown to be calculated by subtracting the energies of the components from the interfacial model, namely,

$$E_{ads} = E_{(adsorbate/substrate)} - E_{adsorbate} - E_{substrate}$$

Where $E_{(adsorbate/substrate)}$, $E_{adsorbate}$, $E_{substrate}$ are the total energies of the adsorbate/substrate system, isolated adsorbate, and substrate.

The Gibbs free energy change (ΔG) ($\Delta G = \Delta H + \Delta E_{ZPE} - T\Delta S$) is taken into account for the gaseous molecules with surfaces, including thermodynamic zero-point energy (ZPE) corrections and entropy contributions derived from standard partition functions or experimental values.^[21, 22]

Absorption enthalpy of NH_3 , NO , H_2O on the surface of Dimeric vanadia model is -22.19 kJ/mol, -10.6 kJ/mol, -47.28 kJ/mol, which is consistent with previously reported values.^[23,24] The results that water adsorption has a lower enthalpy than NH_3 are in agreement with previous studies (Yin et al., 1999; Song et al., 2018; Ranea et al., 2000; Yin et al., 1999).^[25-28]

References:

21. Wei X, Ke Q, Cheng H, et al. Seed-assisted synthesis of Cu-(Mn)-UZM-9 zeolite as excellent NO removal and N_2O inhibition catalysts in wider temperature window. *Chemical Engineering Journal*, 2020, 391: 123491.
22. Jin Q, Chen M, Tao X, et al. Component synergistic catalysis of Ce-Sn-W-Ba- O_x/TiO_2 in selective catalytic reduction of NO with ammonia. *Applied Surface Science*, 2020, 512: 145757.
23. Anstrom M, Dumesic J A, Topsøe N Y. Theoretical insight into the nature of ammonia adsorption on vanadia-based catalysts for SCR reaction. *Catalysis letters*,

2002, 78: 281-289.

24. Smak T Z, Dumesic J A, Clausen B S, et al. Temperature-Programmed Desorption/Reaction and in Situ Spectroscopic Studies of Vanadia/Titania for Catalytic Reduction of Nitric Oxide. *J. Catal*, 1992, 135: 246.

25. Yin X, Han H, Gunji I, et al. NH₃ adsorption on the Brønsted and Lewis acid sites of V₂O₅ (010): a periodic density functional study. *The Journal of Physical Chemistry B*, 1999, 103(22): 4701-4706.

26. Song I, Lee J, Lee G, et al. Chemisorption of NH₃ on monomeric vanadium oxide supported on anatase TiO₂: a combined DRIFT and DFT study. *The Journal of Physical Chemistry C*, 2018, 122(29): 16674-16682.

27. Ranea V A, Vicente J L, Mola E E, et al. Adsorption of H₂O on the (001) plane of V₂O₅: chemisorption site identification. *Surface Science*, 2000, 463(2): 115-124.

28. Yin X, Fahmi A, Han H, et al. Adsorption of H₂O on the V₂O₅ (010) surface studied by periodic density functional calculations. *The Journal of Physical Chemistry B*, 1999, 103(16): 3218-3224.

Comment 1-7. Overall, the mechanism is quite unclear- the author mentioned that “the monomeric surface vanadia site followed a similar reaction pathway except that adsorbed NH₃ interacted with gaseous NO as compared to the dimeric surface. On what basis author conclude this? No clear explanation is given; for example, what are the formation energies of important intermediate species such as nitrate?

Response 1-7:

Thanks for your suggestion.

The calculated formation energy of nitrate species on monomeric and dimeric surface vanadia sites are -0.34 eV and -0.47 eV (Figure S32), respectively, is consistent with the findings of Langhammer D. et al. [22] MES-DRIFTS have detected nitrate species on polymeric vanadia sites, whereas none were observed on monomeric vanadia sites. This indicates that the adsorption of nitrate species is more prone to occur on dimeric surface vanadia sites. The manuscript has been revised to incorporate the updated wording in **lines 372-375**.

Monomeric -0.34 eV

Dimeric -0.47eV

Figure S32 Nitrate formation energy on a) monomeric and b) dimeric vanadia sites.

References:

22. Langhammer D, Kullgren J, Osterlund L. Adsorption and Oxidation of NO₂ on Anatase TiO₂: Concerted Nitrate Interaction and Photon-Stimulated Reaction. *ACS Catalysis*, 2022, 12(16): 10472-10481.

Comment 1-8. Given the intricacy of SCR reactions, can you elaborate on plans for more extensive mechanistic studies to enhance understanding, especially regarding the suggested N-H cleavage and its consequences on surface species? For instance, is there consideration for exploring the potential formation of nitrosamine (N(=O)-NH_2) species as a result of the proposed N-H cleavage?

Response 1-8:

Thanks for your suggestion. We computed the $-\text{NH}_2$ reaction pathway, as depicted in Figure S31. Specifically, the adsorbed L-NH_3 undergoes cleavage to form $-\text{NH}_2$, which, upon the introduction of NO , leads to the species $-\text{NH}_2\text{NO}$. Further decomposition of NH_2NO into N_2 and H_2O completes the reaction cycle. We found that the activation energy from B to C in the $-\text{NH}_2$ pathway at dimeric vanadia sites is 2.08 eV, higher than the 1.08 eV from B to TS1 in the nitrate pathway (Figure 1b). Therefore, on dimeric sites, there is a preference for the reaction to proceed through the nitrate pathway. The corresponding expression has been modified in the main text, specifically in **lines 351 to 357**.

Figure S31 a) NH_2 reaction pathway on dimeric vanadia sites. b) Reaction energies of NH_2 pathway and nitrate pathway in the manuscript.

Comment 1-9. Author should be careful about miss typos as can be seen in line (74) “However, as a case of For the commercial..”. Also, some missing ref, for example, in line (250) “In contrast to Marberger et al.'s study, isotope experiment confirmed that the band at 1599 cm⁻¹ should be attributed to nitrate rather than NH₃, suggesting the involvement of nitrate in the reaction”. Also, I don’t see the information in Figure S26 (SI information) as stated by the author in this statement “The lattice dimension was about 11.28Å × 9.94Å × 20.31Å. V/W oxide clusters were loaded on the substrate to form a VWTi catalyst as shown in Figure S26”. Further details need to be added about the chosen modeled systems.

Response 1-9:

Thanks for your correction.

Typographical error: We have corrected the error in **line 75**.

Missing reference: the missing reference has been added in **line 276**.

Incorrect referencing of the supporting figure: In our revision, we provided additional details and information of the modeled systems in **Figure S29 in lines 516-527**.

Reviewer: 2

Comments:

This study presents a plasma-assisted treatment strategy to promote the oligomerization of vanadium species over VWTi catalysts. The activity enhancement mechanism of the oligomeric vanadia-dominated catalyst was characterized through various experimental measurements and DFT calculation. This topic is interesting and important. However, the manuscript must address the following questions before publication in Nature Communications.

Comment 2-1. The plasma treatment resulted in the polymerization of V species. But the reason and the mechanism were not clarified in this manuscript.

Response 2-1:

When plasma interacts with the catalyst surface, a substantial number of particles are injected onto the material surface. In the process of interaction between ions, neutral particles, and the material surface, the kinetic energy of incident particles is transferred to surface atoms through collision cascades.^[13-19] Due to the lower bond energy of V-O bonds (5.96 kJ/mol) compared to Ti-O bonds (6.82 kJ/mol), and the high coordination number (CN=6) octahedral structure of titanium dioxide, thereby enhancing the stability of the crystal. When the absorption energy of V-O bonds in the plasma exceeds their bond energy, the V-O bonds become more susceptible to breaking, resulting in the migration of V atoms to the surface of the catalyst. On the surface, V atoms undergo aggregation through bonding, forming polymeric vanadium oxide, effectively reducing the system's energy.^[20] Therefore, the breaking of V-O bonds and migration of V atoms to topmost surface in the plasma facilitate the aggregation of vanadium oxide, providing the system with a lower energy state. The relevant expression has been added to **lines 229-239** in the manuscript.

References:

13. Ye Z, Zhao L, Nikiforov A, et al. A review of the advances in catalyst modification using nonthermal plasma: Process, Mechanism and Applications. *Adv. Colloid Interface Sci.*, 2022: 102755.
14. Santos A M, Catapan R C, Duarte D A. The potential of non-thermal plasmas in the

preparation of supported metal catalysts for fuel conversion in automotive systems: A literature overview. *Front. Mech. Eng.*, 2020, 6: 42.

15. Liu X, Long H, Hu S, et al. Photocatalytic TiO₂ nanoparticles activated by dielectric barrier discharge plasma assisted ball milling. *J. Nanosci. Nanotechnol.*, 2020, 20(3): 1773-1779.

16. Dou S, Tao L, Wang R, et al. Plasma - assisted synthesis and surface modification of electrode materials for renewable energy. *Adv. Mater.*, 2018, 30(21): 1705850.

17. Zou J J, Liu C J, Zhang Y P. Control of the metal- support interface of NiO-loaded photocatalysts via cold plasma treatment. *Langmuir*, 2006, 22(5): 2334-2339.

18. Li K, Tang X, Yi H, et al. Research on manganese oxide catalysts surface pretreated with non-thermal plasma for NO catalytic oxidation capacity enhancement. *Appl. Surf. Sci.*, 2013, 264: 557-562.

19. Mistry H, Varela A S, Bonifacio C S, et al. Highly selective plasma-activated copper catalysts for carbon dioxide reduction to ethylene. *Nat. Commun.*, 2016, 7(1): 12123.

20. El-Roz M, Lakiss L, Telegeiev I, et al. High-visible-light photoactivity of plasma-promoted vanadium clusters on nanozeolites for partial photooxidation of methanol. *ACS Appl. Mater. Inter.*, 2017, 9(21): 17846-17855.

Comment 2-2. Line 174-175: With longer treatment time, the reaction rate decreased. Why? Are there any changes in the vanadyl species with the change of treatment time? The authors did not provide an explanation or discussion.

Response 2-2:

To delve into the reasons behind the decline in activity due to prolonged plasma treatment, we analyzed the spectral characteristics and reducibility of the PL-300s catalyst. In the Raman spectra (**Figure S36 a**), PL-300s exhibits a significant reduction in terminal V=O bonds. Further analysis of V 2p XPS (see **Figure S36 b**) reveals a lower energy shift in the valence state of V in the PL-300s sample, indicating a change in the oxidation state of V from 5+ to 4+. This suggests that excessively prolonged plasma treatment can disrupt the structure of surface-active components. ^[21-23]

Through H₂-TPR profiles (**Figure S36 c**), we observe a decrease in hydrogen consumption for the V reduction peak from 1.16 cm³/g in PL to 0.63 cm³/g in PL-300s, indicating a suppression of the redox ability of V. From in-situ UV-vis experiments (**Figure S36 d**), it is evident that after the introduction of NH₃ and NO, the reducible V⁵⁺ content in PL-300s decreases to 11.9%. This indicates a reduction in the number of active V sites. As V=O serves as redox sites in the NH₃-SCR catalytic reaction, activating NH₃ and facilitating the generation of NH₂NO (steps B→C, E→F, J→K, M→N). ^[24-27] Therefore, the deterioration of the NH₃-SCR activity of the catalyst is exacerbated by the prolonged plasma treatment. The analysis has been added to **lines 395-406** in the manuscript.

References:

21. Gong Z, Zhong W, He Z, et al. Regulating surface oxygen species on copper (I) oxides via plasma treatment for effective reduction of nitrate to ammonia. *Appl. Catal. B*, 2022, 305: 121021.
22. Chen M, Chu W, Zhu J, et al. Plasma assisted preparation of cobalt catalysts by sol-gel method for methane combustion. *J. Solgel. Sci. Technol.*, 2008, 47: 354-359.
23. Kong L, Wang C, Zheng H, et al. Defect-induced yellow color in Nb-doped TiO₂ and its impact on visible-light photocatalysis. *J. Phys. Chem. C*, 2015, 119(29): 16623-16632.

24. Rossi L, Palacio M, Villabrille P I, et al. V-doped TiO₂ photocatalysts and their application to pollutant degradation. *Environ. Sci. Pollut. Res.*, 2021, 28: 24112-24123.
25. Jaegers N R, Lai J K, He Y, et al. Mechanism by which tungsten oxide promotes the activity of supported V₂O₅/TiO₂ catalysts for NO_x abatement: structural effects revealed by 51V MAS NMR spectroscopy. *Angew. Chem. Int. Ed.*, 2019, 131(36): 12739-12746.
26. Inomata Y, Kubota H, Hata S, et al. Bulk tungsten-substituted vanadium oxide for low-temperature NO_x removal in the presence of water. *Nat. Commun.*, 2021, 12(1): 557.
27. Zhu M, Lai J K, Tumuluri U, et al. Reaction pathways and kinetics for selective catalytic reduction (SCR) of acidic NO_x emissions from power plants with NH₃. *ACS Catal.*, 2017, 7(12): 8358-8361.

Figure S36 a) Raman spectra of OR, PL and PL-300s. b) V 2p spectra of OR, PL and PL-300s samples. c) H₂-TPR profiles of OR, PL and PL-300s samples. d) Reduction of the surface V⁵⁺ sites for OR, PL, PL-300s and PL-uncalcined samples as a function of environmental conditions. (2000 ppm of NH₃/Ar, 2000ppm of NO+2000ppm of NH₃/Ar in sequence).

Comment 2-3. In Figure 2, PL presented higher ratio of polymeric V species and lower H₂ reduction temperature. It is contradictory. In general, the monomeric V can be reduced by H₂ at lower temperatures.

Response 2-3:

As the reviewer rightly highlighted, there is a prevailing belief that highly polymeric oxide species are challenging to reduce. However, it is worth noting that after plasma modification, the content of vanadium on the topmost surface of the catalyst increases (from 3 at% to 28%). The reducibility of surface vanadium on TiO₂ surpasses that of inner-layer vanadium, and this promotion effect counteracts the negative effect due to agglomeration [28, 29]. This could be the reason for the increased reducibility of PL catalyst and the resulting shift in the reduction peak. We incorporated this statement at **lines 164-165** in the manuscript.

References:

28. Haber J. Fifty years of my romance with vanadium oxide catalysts. *Catal. Today*, 2009, 142(3-4): 100-113.
29. Rossi L, Palacio M, Villabrille P I, et al. V-doped TiO₂ photocatalysts and their application to pollutant degradation. *Environ. Sci. Pollut. Res.*, 2021, 28: 24112-24123.

Comment 2-4. In Line 138, the PL catalyst also exhibited a broad band centered at 930 cm^{-1} assigned to the V-O vibration from the bridging V-O-Ti bond. However, it is different from the Figure 2, which show V-O-V structure.

Response 2-4:

We appreciate the reviewer's astute observation. There is inconsistency in the assignment of the 930 cm^{-1} peak in two locations within the manuscript. Previous reports indicated that the Raman vibrational mode of the V-O-V bond is located at 800 cm^{-1} , while the V-O-Ti vibration occurs at 930 cm^{-1} .^[30-34] Lv Z et al. conducted Raman spectroscopy studies over V_2O_5 , V/Ti, and Ti/V catalysts, they find all the Ti-containing catalysts possessed a feature peak assignable to V-O-Ti (928 cm^{-1}) while this peak was hardly observed over V_2O_5 .^[35] Therefore, this vibration should be attributed to V-O-Ti vibrations rather than V-O-V. Correction has been made in **Fig. 2c** to avoid this misunderstanding.

References:

30. Zhu M, Lai J K, Tumuluri U, et al. Nature of active sites and surface intermediates during SCR of NO with NH_3 by supported $\text{V}_2\text{O}_5\text{-WO}_3/\text{TiO}_2$ catalysts. *J. Am. Chem. Soc.*, 2017, 139(44): 15624-15627.
31. Lai J K, Jaegers N R, Lis B M, et al. Structure–Activity Relationships of Hydrothermally Aged Titania-Supported Vanadium–Tungsten Oxide Catalysts for SCR of NO_x Emissions with NH_3 . *ACS Catal.*, 2021, 11(19): 12096-12111.
32. Lai J K, Wachs I E. A perspective on the selective catalytic reduction (SCR) of NO with NH_3 by supported $\text{V}_2\text{O}_5\text{-WO}_3/\text{TiO}_2$ catalysts. *ACS Catal.*, 2018, 8(7): 6537-6551.
33. He G, Lian Z, Yu Y, et al. Polymeric vanadyl species determine the low-temperature activity of V-based catalysts for the SCR of NO_x with NH_3 . *Sci. Adv.*, 2018, 4(11): eaau4637.
34. Christodoulakis A, Machli M, Lemonidou A A, et al. Molecular structure and reactivity of vanadia-based catalysts for propane oxidative dehydrogenation studied by in situ Raman spectroscopy and catalytic activity measurements. *Journal of Catalysis*, 2004, 222(2): 293-306.

35. Lv Z, He G, Zhang W, et al. Interface sites on vanadia-based catalysts are highly active for NO_x removal under realistic conditions. *J. Environ. Sci.*, 2024, 136: 523-536.

Fig. 2.c *In situ* dehydrated Raman spectra of PL (blue) and OR (red).

Comment 2-5. In Figure 2d, the fitting of NMR was unreasonable for the noticeably different FWHM between the two catalysts.

Response 2-5:

Thank you for bringing this to our attention. Upon a thorough reassessment, we acknowledge that the fitting of NMR may not sufficiently represent the distinctive variations in full width at half maximum (FWHM) observed in the depicted spectra. In response, we have conducted a comprehensive re-peaking of the NMR spectra, ensuring consistency in FWHM, and we have incorporated the updated NMR images in the **Fig.2d** of the manuscript.

Fig. 2.d Deconvolution of the *in situ* solid state ^{51}V MAS NMR spectra of PL (blue) and OR (red)

Comment 2-6. Line 231: The peak at 1330 cm⁻¹ was attributed to NH₂NO species, which is inconsistent with the literature. Previous studies have found that the characteristic peak of NH₂NO is at 1490 cm⁻¹ over VWTi catalysts (Angew. Chem., Int. Ed. 2016, 55, 11989)

Response 2-6:

Thank you for bringing this to our attention. Through further analysis of MES-DRIFTS in Fig.4b, we have identified a peak at 1490 cm⁻¹ that aligns with the introduction of NO, and according to Marberger et al., this is attributed to the $\nu(\text{N}=\text{O})$ vibration in NH₂NO.^[12] We have incorporated the assignment of this peak into **Fig. 4b** and the manuscript at **line 251**.

Additionally, evidence from previous studies suggests attributing the 1330 cm⁻¹ peak to the N-H vibration of NH₂NO.^[36-38] In **Fig.4b**, the phase of 1330 cm⁻¹ peak aligns with nitrate species associated with NO introduction, contrasting with species related to NH₃, such as L-NH₃ and B-NH₃, which exhibit an opposite phase. This indicates that the presence of this species is a result of the introduction of NO. In **Fig. 4c**, MES-DRIFTS experiments with ND₃/NO revealed the absence of an absorption peak at 1330 cm⁻¹, indicating its association with N-H vibration. Therefore, we attribute the peak at 1330 cm⁻¹ to the NH₂ symmetric stretching vibration in NH₂NO.

References:

36. Zhao Q, Chen B, Li J, et al. Insights into the structure-activity relationships of highly efficient CoMn oxides for the low temperature NH₃-SCR of NO_x. *Appl. Catal. B*, 2020, 277: 119215.
37. Wu X, Yu W, Si Z, et al. Chemical deactivation of V₂O₅-WO₃/TiO₂ SCR catalyst by combined effect of potassium and chloride. *Front. Environ. Sci. Eng.*, 2013, 7: 420-427.
38. Jacox M E, Thompson W E. Infrared spectra of NH₂NO, NH₂NO⁺, and NNOH⁺ and of the N₂···H₂O complex trapped in solid neon. *J. Chem. Phys.*, 2005, 123(6).

Fig 4.b MES DRIFT spectra of PL during NO+O₂/NH₃+O₂ modulation experiment.

Comment 2-7. Lines 266-270: L-NH₃ (i.e., NH₃ adsorbed on surface metal sites) is considered as the primary active species involved in the SCR reaction, which is inconsistent with the adsorption sites of NH₃ in the DFT calculation models.

Response 2-7:

Thanks for your suggestion.

We calculated the structure X of L-NH₃ adsorbed on V, as shown in **Figure S31**. We found that the energy of structure X is higher than that of structure B. Eventually, L-NH₃ on structure X migrates to the surface and transforms into structure B. Therefore, structure B is identified as the most stable configuration, with NH₃ adsorbed on the catalyst surface, participating in the NH₃-SCR reaction. The relevant expression has been added in the manuscript, specifically in **lines 359-362**.

Figure S31 a) L-NH₃ pathway on dimeric vanadia sites and b) corresponding reaction energies.

Comment 2-8. Line 328: The dehydration process is the rate-determining step, which is unbelievable. If the effect of entropy is considered (i.e., calculating the free energy), the desorption energy of water molecules would be much smaller.

Response 2-8:

Thanks for your suggestion. We calculated the intermediate species (IS1 and IS2) between the G→H and N→O steps (**Figure S33**). The energy barriers for the dehydration steps from IS1 to H and IS2 to O are 0.91 and -1.38 eV, respectively, both lower than the activation energy of 1.17 eV for the transition from E to TS2. Thus, the dehydrogenation step leading to the generation of TS2 in this reaction cycle remains the rate-determining step with the highest activation energy. The pertinent statement has been incorporated into the manuscript, precisely in **lines 382-385**.

Figure S33 a) G-IS1-H and N-IS2-O pathway on dimeric vanadia sites and b) corresponding reaction energies.

Comment 2-9. DFT calculation method section: The thickness of the vacuum layer, whether the bottom layer of the model is fixed, and the calculation method for the transition state are not defined.

Response 2-9:

The thickness of vacuum layer of anatase (101) surface was set over 15Å. For relaxation of clean anatase surface, atoms at bottom four layer were fixed. For relaxation of vanadia loaded anatase surface and gases absorbed models, atoms at bottom eight layers were fixed, which means the upper vanadia clusters were allowed to relax and interact with gas molecules. This section has been incorporated into the manuscript in the description of DFT calculations, specifically in **lines 516-527**.

Comment 2-10. Figure 5a: The transition states for the generation of H₂O (G→H and N→O processes) have not been provided. In fact, the two -OH groups are so far apart that H cannot directly transfer to form one water molecule.

Response 2-10:

Thanks for your suggestion. We calculated the intermediate species (IS1 and IS2) between the G→H and N→O steps, and their energy changes are shown in **Figure S33**. The activation energies from G to IS1 and N to IS2 are -0.49 eV and -2.72 eV, respectively, both lower than the activation energy of 1.17 eV for TS2. The relevant expression has been added in the manuscript, specifically in **lines 382-385**.

Reviewer: 3

Comments:

The authors successfully demonstrated the preparation of $V_2O_5-WO_3/TiO_2$ by plasma treatment and its enhanced activity in NH_3 -SCR. Many characterization results have revealed the state of $V_2O_5-WO_3/TiO_2$, and detailed studies have been conducted. However, some improvements are needed to facilitate reader understanding. Detailed comments are given below.

Comment 3-1. In the same sentence on line 74, 'however' is used twice.

Response 3-1:

We appreciate the reviewer's keen observation. In **line 75**, the error has been rectified in the manuscript. Thank you for bringing this to my attention.

Comment 3-2. Fig. 1a is not mentioned in the text. Please add it to the text.

Response 3-2:

Thank you for your valuable suggestion. We have incorporated a description of the preparation process schematic mentioned in Fig. 1a into the manuscript in **line 86**.

Comment 3-3. Fig. S6 is PL for both.

Response 3-3:

Thank you for your suggestion. **Figure S6** a) corresponds to PL, and b) corresponds to OR. We have made correction in the caption of **Figure S6**.

Figure S6 XPS O1s spectra of PL a) and OR b).

Comment 3-4. Please explain which color is the surface side in Fig. S7 and S8

Response 3-4:

Thank you for your suggestion. We have added markings to distinguish curves representing the surface and inner layers in **Figure S7 and S8**.

Figure S7 HS-LEIS spectral of PL

Figure S8 HS-LEIS spectral of OR

Comment 3-5. In line 170, the value of GHSV is different from SI. Also, please provide the actual flow rate, since some readers may refer to the actual flow rate.

Response 3-5:

Thank you for your advice. Upon our review, we found a typographical error, and the correct GHSV is 37500. We have made the correction in the supporting information in **line 18.**

Comment 3-6. In line 200, indicate which sample was calcined.

Response 3-6:

Thank you for your valuable suggestion. We acknowledge that our expression was not clear enough. The samples subjected to Re-calcination are abbreviated as RC, and we have added this clarification in the captions of **Table S3** and **Figure S13**.

Comment 3-7. In lines 202-207, a control experiment is performed, but the sample names are abbreviated in the SI. Please explain which sample corresponds to which sample name.

Response 3-7:

Thank you for your guidance, and we appreciate your attention to detail. We did not provide sample abbreviations in the main text, and to avoid any potential misunderstanding, we have now included annotations in the manuscript in **lines 211-215**.

Comment 3-8. The term 'carrier' is used in line 212, whereas 'support' is usually used in this field.

Response 3-8:

Thanks for your advice. We have made the correction in **line 220**.

REVIEWER COMMENTS

Reviewer #1 (Remarks to the Author):

I'm happy to accept the article since the authors have carefully handled my comments.

Reviewer #2 (Remarks to the Author):

Some of my questions have been addressed, but there are still several unresolved issues, which significantly affect the quality of the article.

(1) Response to Comment 2-1: The authors propose that the bond energies of Ti-O and V-O bonds are 6.82 kJ/mol and 5.96 kJ/mol, respectively. Where do the data come from? The bond energy of a typical chemical bond should be several hundred kJ/mol.

(2) Response to Comment 2-8 and Comment 2-10: The author's understanding of energy barrier/activation energy is incorrect. The energy barrier is the energy difference between the transition state and the initial state, not the energy difference between the intermediate and the initial state. The energy barrier and activation energy cannot be negative. The misunderstandings of energy barriers are likely to affect the correctness of the relative energy profiles of the entire reaction pathway.

(3) Response to Comment 2-9: The calculation method for the transition state is still not provided. The imaginary frequency of the transition state is also not given. The information is crucial for determining the reliability of the calculations.

Reviewer #3 (Remarks to the Author):

The paper has been revised and improved. No questions from me.

**Plasma-Assisted Manipulation of Vanadia**
**Nanoclusters for Efficient Selective Catalytic**
**Reduction of NO_x**

Yong Yin ^{a#}, Bingcheng Luo ^{b#}, Kezhi Li ^c, Benjamin M. Moskowitz ^d, Israel E. Wachs* ^d,
Minghui Zhu* ^e, Tianle Zhu ^a and Xiang Li ^{*a}

6 ^a School of Space and Environment, Beihang University, Beijing, 100191, China.

7 ^b College of Science, China Agricultural University, Beijing 100083, China.

8 ^c Institute of Engineering Technology, Sinopec Catalyst Co., Ltd., Beijing, 101111, P.R.
9 China

10 ^d Operando Molecular Spectroscopy & Catalysis Laboratory, Department of Chemical and
11 Biomolecular Engineering, Lehigh University, Bethlehem, Pennsylvania 18015, United
12 States.

13 ^e State Key Laboratory of Chemical Engineering, East China University of Science and
14 Technology, 130 Meilong Road, Shanghai 200237, China.

# These authors contributed equally to this work.

*Corresponding author's E-mail address: iew0@lehigh.edu (I. E. Wachs), minghuizhu@ecust.edu.cn
(M.Zhu) and xiangli@buaa.edu.cn (X. Li).

Tel.: +86 10 82314215

**Abstract**

[revised manuscript text omitted]
 (ν_s (N-O)₂ at 1288 cm⁻¹ and ν_{as} (N-O)₂ at 1599 cm⁻¹), bidentate nitrate (ν_{as}
(N-O)₂ at 1579 cm⁻¹), and NH₂NO (ν_s (N-H) at 1330 cm⁻¹, ν (N=O) at 1490 cm⁻¹) as shown
in Figure S16b^{18, 53-57}. For the PL catalyst, the temperature for formation of N₂ in the outlet
initiated at 115 °C that was much lower than the initiation temperature of 180 °C for the
OR catalyst reflecting the greater activity for the PL catalyst (Figure S17).

The MES-DRIFTS measurements were conducted at 150°C in order to determine the
participating surface species in the SCR reaction. The MES studies employed alternating
pulses of NH₃ and NO while maintaining a constant O₂ concentration (5 vol%) in a flowing
Ar environment (Details given in Figure S18, S20 and S22). In Fig. 4a and 4b, the PL
catalyst exhibited the MES-DRIFTS peaks from NH₃ related peaks (1190, 1370, 1460,
1542, 1618, 3260, 3406 cm⁻¹) and adsorbed H₂O (1618 cm⁻¹). The V=O (2035 cm⁻¹) at
overtone region showed an opposite sign to NH₃ introduction, indicating NH₃ adsorption
on it (Figure S19). The bridging nitrates (1288 ν_s (N-O)₂ and 1599 cm⁻¹ ν_{as} (N-O)₂), bidentate

[revised manuscript text omitted]

2000ppm of NH_3 + 2000ppm of NO (50 ml/min), and 2000ppm of NH_3 + 2000ppm of NO
(50 ml/min) + 5 % O_2 (50 ml/min). The Raman spectra were recorded every 2min at a
resolution of 2cm^{-1} . The same pulse sequence and data processing methods as FTIR
were employed for the Raman concentration modulation excitation experiments. The
Raman spectra were recorded every 30 s at a resolution of 4cm^{-1} .

**Computational Details**

First-principles calculations were performed using the density functional
theory framework within the Vienna ab initio simulation package (VASP 5.4.4) 77-79. A (3x1)
supercell of the anatase (101) surface with double layer was employed as substrate for
the commercial SCR catalyst surface 80-82. The thickness of vacuum layer of anatase (101)
surface was set over 15\AA . For relaxation of vanadia loaded anatase surface and gases

absorbed models, atoms at bottom eight layers were fixed, which means the upper
vanadia clusters were allowed to relax and interact with gas molecules. After geometric
optimization with GGA-PBE functionals, the lattice parameters became $11.28\text{\AA} \times 9.94\text{\AA} \times$
20.31\AA , which is in good agreement with input experimental lattice parameters ($11.33\text{\AA} \times$
$10.2\text{\AA} \times 20.84\text{\AA}$) of anatase (101) surface shown in Figure S29. Perdew–Burke–
Ernzerhof (PBE) functionals, based on the generalized gradient approximation (GGA),
were widely used to account for exchange–correlation of $\text{V}_2\text{O}_5/\text{TiO}_2$ catalyst for selective
catalytic reduction with ammonia^{22, 82-85}. The interaction between the ions and the
electrons was described by projector augmented wave (PAW) methods⁸⁵. The
pseudopotentials used for the present models were constructed by the electron
configurations as V $3s^2 3p^6 3d^4 4s^1$ states, Ti $3s^2 3p^6 3d^2 4s^2$ states, N $2s^2 2p^3$ states, H $1s$
states, and O $2s^2 2p^4$ states. The energy cut-off value was set at 600 eV⁸⁶. The
convergence criteria of total energies and forces were 10^{-6} eV/atom and 0.05 eV/Å. The
first Brillouin zone was sampled by a Monkhorst-Pack $2 \times 2 \times 1$ K-point mesh⁸⁷. The
adsorption energies and electron density difference were calculated according to the
adsorption or interfacial models⁸⁸⁻⁹¹. We used dimeric vanadyl species as the model for
our DFT calculations because they are the basic structural unit of various polymeric
vanadia structures and can reasonably represent the coupling effect in them. The coupling
effect between two adjacent vanadyl species (i.e., within a dimer unit of vanadia) at the
reaction site was common in dimeric and higher-order polymeric vanadia structures. It
sped up the whole catalytic cycle during the NH_3 -SCR of NO over the polymeric vanadyl
species, and thus, we expected that dimeric and higher-order polymeric vanadia would
have similar effects on the SCR reaction. Free energy correction was performed by
including the zero-point energy (ZPE) and enthalpic and entropic contributions from
vibrational degrees of freedom, with the substrate fixed. Climbing Image Nudged Elastic
Band (CI-NEB) method was employed to find the minimum energy path (MEP) connecting
the reactants and products⁹¹⁻⁹³. The fast inertial relaxation engine (FIRE) was used as
optimizer in CI-NEB.

**Data availability**

The datasets generated during and/or analyzed during the current study are available
from the corresponding authors upon reasonable request. NMR, FTIR, Raman, UV-vis
spectra and HS-LEIS data that support the findings of this study are available in

[revised manuscript text omitted]

**Additional Information**

**Supplementary information** The online version contains supplementary material
available at https://doi.org/10.1038/s***-**-*-x

**Acknowledgments**

This work was financially supported by the National Natural Science Foundation of
China (22176008, 21906004, and 52202154). Bingcheng Luo acknowledges support from
the High-performance Computing Platform of China Agricultural University. We thank Si
Jiang for the scientific discussion and valuable suggestions. We thank Didi Li at East China
University of Science and Technology for their help with X-ray absorption spectroscopy
characterization. We thank Yuan Xu (Bruker Co.) for assistance with collecting Raman
spectra. We acknowledge support from National Supercomputer Center in Tianjin, and the
energy calculations were performed on Tianhe new generation supercomputer. We thank
Xiumei Wang (Bruker NMR Facility) for assistance with collecting NMR spectra. The work
at Lehigh University was supported as part of Understanding & Control of Acid Gas-
Induced Evolution of Materials for Energy (UNCAGE-ME), an Energy Frontier Research
Center funded by the U.S. Department of Energy, Office of Science, Basic Energy
Sciences under Award # DE-SC0012577.

**Competing interests**

The authors declare no competing interests.

Figure Captions

Fig.1 | Synthesis and electron microscopy of the samples. **a** Schematic of the surface vanadyl species under plasma treatment over vanadia-based catalysts. HRTEM images of **b** OR and **c** PL. Selected high-angle annular dark field (HAADF) images of **d** OR and **e** PL. Enlarged view of the yellow region in **d** (**f**) and **e** (**g**). 2D atomic maps of the EELS signals of **h** Ti, **i** O and **j** V in combination with the simultaneously acquired HAADF image of red region in **g**.

Fig.2 | Spectral Characterization of the samples. **a** V2p and W4f XPS spectra, **b** H₂-TPR profiles of PL (blue) and OR (red). **c** *In situ* Raman spectra, and **d** Deconvolution of the *in situ* solid state ⁵¹V MAS NMR spectra of PL (blue) and OR (red).

Fig.3 | Catalytic performance of PL and OR catalyst in the NH₃-SCR. **a** Reaction rate and N₂ selectivity of PL with different treatment time at 200 °C for NH₃-SCR. **b** Reaction rate and apparent activating energy (E_a, inset) of PL and OR. **c** NO/NH₃-SCR cycle stability of OR (orange) and PL (blue). Comparison of TOF values for the **d** reported NO/NH₃-SCR catalysts and **e** different vanadia-based (PL and OR) NO/NH₃-SCR catalysts at 200 °C.

Fig.4 | MES DRIFT, Raman, and in-situ Raman spectra on PL and OR catalyst. **a** The diagram of surface species corresponding to IR vibrations. MES DRIFT spectra during **b** NO+O₂/NH₃+O₂ and **c** NO+O₂/ND₃+O₂ modulation experiment. **d** *in situ* Raman spectra of OR and PL under different reaction conditions (2000 ppm of NH₃/Ar, 2000ppm of NO+2000ppm of NH₃/Ar, 2000ppm of NO+2000ppm of NH₃+5% O₂/Ar in sequence). **e** MES Raman spectra of PL during NO+O₂/NH₃+O₂ modulation experiment. The above experiments were carried out at 150°C.

Fig.5 | DFT calculations for the reaction pathway of SCR over dimeric and monomeric vanadia. The optimized molecular structures for the reactant, transition states, intermediates, product, and reaction energies were determined using DFT theory for each elementary step in the NH₃-SCR mechanism over the surfaces of **a** dimeric vanadia and **b** monomeric vanadia. Red, cyan, green, blue, and white circles denote O, Ti, V, N, and H atoms, respectively.

Fig. 1: Synthesis and electron microscopy of the samples.

a Schematic of the surface vanadyl species under plasma treatment over vanadia-based catalysts. HRTEM images of **b** OR and **c** PL. Selected high-angle annular dark field (HAADF) images of **d** OR and **e** PL. Enlarged view of the yellow region in **d** (**f**) and **e** (**g**). 2D atomic maps of the EELS signals of **h** Ti, **i** O and **j** V and W in combination with the simultaneously acquired HAADF image of the red region in **g**.

Fig. 2: Spectral Characterization of the samples.

a V2p and W4f XPS spectra, b H₂-TPR profiles of PL (blue) and OR (red). c *In situ* dehydrated Raman spectra, and d Deconvolution of the *in*
 *situ* solid state ⁵¹V MAS NMR spectra of PL (blue) and OR (red).

Fig. 3: Catalytic performance of PL and OR catalyst in the NH₃-SCR.

a Reaction rate and N₂ selectivity of PL with different treatment time at 200 °C for NH₃-SCR. b Reaction rate and apparent activating energy

(Ea,inset) of PL and OR. **c** NO/NH₃-SCR cycle stability of OR (orange) and PL (blue). Comparison of TOF values for the **d** reported NO/NH₃-SCR
catalysts and **e** different vanadia-based (PL and OR) NO/NH₃-SCR catalysts at 200 °C.

Fig. 4 MES DRIFT, Raman and *in-situ* Raman spectra on PL and OR catalyst.

**a** Schematic of surface species and their corresponding IR vibrations. MES DRIFT spectra of PL during **b** NO+O₂/NH₃+O₂ and **c** NO+O₂/ND₃+O₂
modulation experiment. **d** *in situ* Raman spectra of OR and PL under different reaction conditions (5% O₂/Ar, 2000 ppm of NH₃/Ar, 2000ppm of
NO+2000ppm of NH₃/Ar, 2000ppm of NO+2000ppm of NH₃ +5% O₂/Ar in sequence). **e** MES Raman spectra of PL during NO+O₂/NH₃+O₂
modulation experiment. The above experiments were carried out at 150°C.

Fig. 5 DFT calculations for the reaction pathway of SCR over dimeric and monomeric vanadia.

The optimized molecular structures for the reactant, transition states, intermediates, product, and reaction energies were determined using DFT theory for each elementary step in the NH₃-SCR mechanism over the surfaces of **a** dimeric surface vanadia site and **b** monomeric surface vanadia

site. Red, cyan, green, blue, and white circles denote O, Ti, V, N, and H atoms, respectively.

REVIEWERS' COMMENTS

Reviewer #2 (Remarks to the Author):

The author has responded well to my concerns, and now I have no further questions.